



# Biomass burning smoke heights over the Amazon observed from space

Laura Gonzalez-Alonso[1], Maria Val Martin[1,2], and Ralph A. Kahn[3]

[1]Chemical and Biological Engineering Department, University of Sheffield, UK
[2]Now at Leverhulme Centre for Climate Change Mitigation, Animal Plant Sciences Department, University of Sheffield, UK
[3]Climate and Radiation Laboratory, Code 613, NASA Goddard Space Flight Center, USA

**Correspondence:** Laura Gonzalez-Alonso (lgonzalezalonso1@sheffield.ac.uk)

**Abstract.** We characterise the vertical distribution of biomass burning emissions across the Amazon during the biomass burning season with an extensive climatology of smoke plumes derived from MISR and MODIS (2005–2012) and CALIOP (2006–2012) observations. Smoke plume heights exhibit substantial variability, spanning a few hundred meters up to 6 km above the terrain. However, the majority of the smoke is located at altitudes below 2.5 km. About 60% of smoke plumes are observed during drought years, at the peak month of the burning season (September; 40–50%) and over tropical forest and savanna regions (94%). At the time of the MISR observations (10:00–11:00 LT), the highest plumes are detected over grassland fires (∼1100 m maximum plume height average) and the lowest plumes occur over tropical forest fires (∼800 m). A similar pattern is found later in the day (14:00–15:00 LT) with CALIOP, although at higher altitudes (2300 m grassland versus 2000 m tropical forest), as CALIOP typically detects smoke at higher altitudes due to its greater sensitivity to thin aerosol layers. On average, 3–20% of the fires inject smoke into the free troposphere; this percentage can increase toward the end of the burning season (November; 15–40%). We find a well-defined seasonal cycle between MISR plume heights, MODIS Fire Radiative Power (FRP) and atmospheric stability across the main biomes of the Amazon, with higher smoke plumes, more intense fires and reduced atmospheric stability conditions toward the end of the burning season. Lower smoke plume heights are detected during drought (800 m) compared to non-drought (1100 m) conditions, in particular over tropical forest and savanna fires. Drought conditions favours understory fires over tropical forest, which tend to produce smouldering combustion and low smoke injection heights. Droughts also seem to favour deeper boundary layers and the percentage of smoke plumes that reach the FT is lower during these dry conditions. Consistent with previous studies, the MISR mid-visible aerosol optical depth demonstrates that smoke makes a significant contribution to the total aerosol loading over the Amazon, with important implications for air quality. This work highlights the importance of biome type, fire properties and atmospheric conditions for plume dynamics, as well as the effect of drought conditions on smoke loading. In addition, our study demonstrates that combined observations of MISR and CALIOP allows for better constraints on the vertical distribution of smoke from biomass burning over the Amazon.

## 1 Introduction

Fires burn across the Amazon region every year, releasing large amounts of trace gases and aerosols into the atmosphere (eg Andreae and Merlet, 2001). The majority of these fires are of anthropogenic origin, e.g., deforestation, preparation of



agriculture fields, conversion of cropland to pasture or road and city expansion (Cochrane, 2003). Between 1976 and 2010, deforestation fires have destroyed more than 15% of the original Amazonian forest (Aragao et al., 2014). Most of these fires burn along the so-called arc of deforestation during the dry season (typically from July to November) (Malhi et al., 2008). However, significant variability exists, caused by changes in meteorology, drought and land-management policies (eg, Van der

Werf et al., 2010; Alencar et al., 2011; Nepstad et al., 2006). Amazon fires can contribute up to about 15% of the total global biomass burning emissions (Van der Werf et al., 2010). These emissions have important implications for air quality, atmospheric composition, climate and ecosystem health (eg, Johnston et al., 2012; Ramanathan et al., 2001; Pacifico et al., 2015). For example, air pollution from deforestation fires is estimated to cause on an average about 3,000 premature deaths per year across South America (Reddington et al., 2015) and may decrease the net primary productivity in the Amazon forest as a

result of increases in surface ozone (Pacifico et al., 2015).

Fires are also an important source of buoyancy locally, which in combination with atmospheric conditions determines the vertical distribution of fire emissions in the atmosphere near the fire source [i.e., injection height]. The altitude to which smoke is injected is critical, as it determines the lifetime of the pollutant, its downwind transport dispersion pathway, and the magnitude of its environmental impact. Space-borne observations have been used to study smoke injection heights across

the world. Using Multi-angle Imaging Spectro Radiometer (MISR) stereo-height retrievals, smoke plume heights have been assessed across North America (Kahn et al., 2008; Val Martin et al., 2010), Indonesia (Tosca et al., 2011), Australia (Mims et al., 2010), southeast Asia (Jian and Fu, 2014), and Europe (Sofiev et al., 2013). For example, Val Martin et al. (2010), using a 5-year climatology of smoke fire plumes and smoke clouds observed by MISR across North America, showed that wildfire smoke can reach altitudes from a few hundred meter above the ground to about 5 km, and that 5–30% of the smoke plumes

are injected into the free troposphere (FT), depending on the biome and year. Related work also demonstrated the important effect that fire intensity and atmospheric conditions have on the initial rise of fire emissions (Kahn et al., 2007; Val Martin et al., 2012). Tosca et al. (2011) reported that less than 4% of smoke plumes reach the free troposphere, based on a MISR 8-year climatology from tropical forest and peatland fires over Borneo and Sumatra, and found that highest plume heights were recorded during an El Niño year over Borneo.

Smoke plume heights have also been determined using space-borne lidar observations from CALIOP (Labonne et al., 2007; Huang et al., 2015), aerosol index from the TOMS and OMI instruments (Guan et al., 2010), and CO observations from TES and MSL (Gonzi and Palmer, 2010). Huang et al. (2015) used a multi-year record of CALIOP vertical aerosol distributions to study smoke and dust layer heights over six high-aerosol-loading regions across the globe. Specifically over the Amazon, they found that on a broad scale, smoke layers are typically located above boundary layer clouds, at altitudes of 1.6–2.5 km.

Consistent with the smoke altitudes detected by CALIOP, an analysis of injection heights using CO observations from TES and MLS estimated that about 17% of fire plumes over South America reached the free troposphere in 2006 (Gonzi and Palmer, 2010).

Numerous studies have sought to understand the impact of biomass burning in the Amazon, from local to hemispheric scales. In particular, during the past decade, several aircraft campaigns have been designed to study the effect of biomass burning on

greenhouse gases, aerosols loading, clouds, regional weather and/or climate over the Amazon [e.g., BARCA (Andreae et al.,



2012), SAMBBA (Allan et al., 2014) and GoAmazon (Martin et al., 2016)]. For example, multiple observations collected during SAMBBA showed that aerosols from biomass burning modify local weather (Kolusu et al., 2015) and regional climate (Thornhill et al., 2017) by reducing cloud cover due to decreased deep convection, stabilizing the boundary layer and suppressing surface fluxes. Based on lidar observations taken in six research flights during SAMBBA (September 16–29, 2014),

Marenco et al. (2016) reported the presence of two distinct smoke aerosol layers, a fresh smoke layer extending from the surface to an altitude of 1–1.5 km, and an elevated and persistent layer of aged smoke at 4–6 km. During the 2008 dry biomass season, continuous raman lidar measurements of optical properties taken in Manaus (2.5° S, 60° W) also detected biomass burning layers at 3-5 km heights, although most of smoke was confined below 2 km (Baars et al., 2012). Whilst the results from these aircraft observations are significant, there are no analyses yet that seek to quantify the vertical distribution of smoke

from fires across the Amazon, and to identify the key factors that control plume rise over this region.

We present here an 8-year climatology of smoke plume heights over the Amazon, derived from observations by the MISR and CALIOP instruments on board the NASA Terra and CALIPSO satellites, respectively. These data are analysed in combination with measurements of fire radiative power (FRP) from NASA's MODerate resolution Imaging Spectroradiometer (MODIS) instruments, assimilated meteorological observations from MERRA-2 and drought condition indicators from the

MODIS Drought Severity Index (DSI). The objectives of this work are to characterise the magnitude and variability of smoke heights from biomass burning across the Amazon, and to assess the influence of fire intensity, local atmospheric conditions, and regional drought on smoke vertical distribution as well as aerosol loading.

## 2   Data and Methods

We use a combination of remote sensing data from multiple sources to build a comprehensive climatology of smoke plume

heights and characterise the vertical distribution of smoke across the Amazon. We provide below a summary of main datasets and tools used in the analysis.

### 2.1   MINX overview

The MISR Interactive Explorer (MINX) software is an application written in Interactive Data Language (IDL) that is used to analyse the physical properties of smoke plumes and to study plume dynamics (Nelson et al., 2013). MINX can use MODIS

thermal anomalies to locate active fires, and MINX then computes the smoke plume or cloud heights from MISR stereo imagery. MINX also collects particle property results from the MISR Standard aerosol retrieval algorithm (Martonchik et al., 2009). MODIS and MISR are both aboard the NASA Terra satellite, which crosses the equator in the descending node at around 10:30 a.m. local time. These instruments allow temporally and spatially coincident detection of active fires and their associated smoke plumes (Kahn et al., 2008).

MODIS has a cross-track swath of 2330 km that provides global coverage every one to two days. The instrument has 36 spectral channels with wavelengths between 0.4 $\mu$m and 14.2 $\mu$m, and detects thermal anomalies at 1 km spatial resolution (at nadir), under cloud-free conditions. MODIS reports fire radiate power based on a detection algorithm that uses brightness





temperature differences in the 4 $\mu$m and the 11 $\mu$m channels (Giglio et al., 2003); this FRP parameter is used as an indicator of fire location and intensity.

MISR has nine push-broom cameras placed at viewing angles spanning -70.5 to 70.5 relative to nadir in the satellite along-track direction (Diner et al., 1998). The cameras each provide imagery in four spectral bands (446, 558, 672, and 867 nm),

which makes it possible to distinguish aerosol types qualitatively (Kahn and Gaitley, 2015) and surface structure from the change in reflectance with angle. This passive stereoscopic imagery method produces cloud and aerosol plume heights, along with cloud-tracked winds aloft. MISR has a swath of 380 km common to all cameras, so global coverage is obtained every nine days at the Equator and every two days at the poles (Diner et al., 1998). The MISR Standard stereo-height product provides vertical resolution of 275–500 meters and horizontal resolution of 1.1 km (Moroney et al., 2002; Muller et al., 2002).

MINX has a graphical user interface that displays the nine MISR multi-angle images. They can be visualised one by one or as an animated loop, providing a 3-D view of the plume that can help in assessing its structure and dynamical behaviour. In addition, MODIS thermal anomalies can be superimposed, which helps identify the locations of smoke sources from active fires. A user needs to digitise the boundaries of the plume, starting at the source point, and to indicate the direction of smoke transport. The MINX stereoscopic algorithm then calculates wind speed from the plume contrast elements, which is used

subsequently to compute wind-corrected heights, accounting for displacement due to the proper motion of the plume elements between camera views. As with the MISR Standard stereo-height product, MINX automatically retrieves smoke plume heights and wind speed at a horizontal resolution of 1.1 km and vertical resolution of 250–500 m, but with greater accuracy for the plume itself, due to the user inputs (Nelson et al., 2013). MINX heights are reported above the geoid, but it also provides local terrain height from a digital elevation map (DEM) product. Here we report height above the terrain, by taking account of the

DEM values. Additional information from the MISR Standard Aerosol product about aerosol amount and type is collected and reported, along with FRP from MODS (Nelson et al., 2013). MINX has been successfully used to investigate fire smoke plume heights over many regions across the world (eg, Kahn et al., 2008; Val Martin et al., 2010; Tosca et al., 2011; Jian and Fu, 2014).

There are several limitations to the MISR-MINX approach that must be considered when studying smoke plume heights.

For example, MISR obtains global coverage only about once per week, and the Terra overpass time in late morning does not coincide with the typical, late-afternoon peak of fire intensity. MODIS does not observe FRP under cloud and dense smoke, and the MINX operator must decide whether to include any pyro-cumulus clouds in the plume-height retrieval. These are the key limitations; they and others are discussed further in the literature (eg, Kahn et al., 2007; Val Martin et al., 2010; Nelson et al., 2013). In addition, three MINX versions were used to generate the data in this study, which might introduce an additional bias.

MINXv2 and v3 included only MISR red-band plume height retrievals, whereas MINXv4 considers both red and blue-band images. Over land, digitalisation with the blue band usually provides higher quality retrievals, especially for optically thin plumes over bright surfaces (Nelson et al., 2013). In contrast, red-band provides higher vertical resolution over dark surfaces and sometimes performs better for optically dense smoke layers. We take these limitations into account throughout our analysis.



## 2.2 MINX smoke plume database

We limited our study to the burning season (July–November) for the period of 2005–2012. Using MINX, we developed a climatology of plume heights across the Amazon, consisting of 10,858 smoke plumes in the region ( 25°S–5°N latitude and 80°W–40°W longitude). Over this domain, the NASA Terra satellite overpass is every 4–8 days at 10:00–11:00 local

time. Table 1 summarises the number of smoke plumes in each year and the digitising source. The climatology includes a combination of smoke plumes extracted from different projects and created with different versions of MINX (v2–4): plumes for August–September in years 2006 and 2007 are from the MISR Plume Height Project (Nelson et al., 2013); plumes in year 2008 are from the global digitalisation effort made for the AeroCom project (MPHP2 and Val Martin and Kahn (2018)); and the five remaining years and additional months are digitised as a part of the current project.

MINX computes several plume heights that describe the altitude that smoke reaches in the atmosphere. In this work, we use the best estimate maximum and median smoke plume heights. In addition, as in previous studies, we remove smoke plumes with poor-quality retrieval flags. This screening leaves a total of 5393 plumes, about 56% of the original database, with 77% and 23% plumes digitised in the red and blue bands, respectively. Our final dataset includes plumes digitised in years with intense fire activity associated to severe drought conditions (e.g., 2005, 2007 and 2010) (Chen et al., 2011),in years with low

fire intensity and considerable precipitation (2009 and 2011) (Marengo et al., 2013) and in one year when land-management policy measures limited deforestation (2006) (Nepstad et al., 2006). Thus, our climatology is intended to capture smoke plumes variability under diverse conditions.

As mentioned in section 2.1, the MISR colour band image used by the MINX algorithm to compute smoke plume heights influences the quality of the plume height and wind speed retrievals. A large majority of the fires detected across our domain

has optically thin smoke plumes. Thus, blue band plume retrievals are more successful, with about 60% of the smoke plumes receiving good or fair quality flags, compared to 36% for the red band retrievals. In our dataset overall, most of plumes are digitised with red band images, as it was the default option for MINX v2–3. However, whenever both band retrievals are available for a plume, blue band is preferred. The choice of the band colour for the retrievals does not affect significantly the results presented here, as the difference in heights for smoke plumes digitised with both bands is negligible (∼60 m), lower

than the ±250 m MINX uncertainty.

## 2.3 Land cover unit data

We use the MODIS Level 3 land cover product MOD12Q1 (Friedl et al., 2010) to determine the type of land cover associated with each of our fire smoke plumes. This product contains 17 International Geosphere-Biosphere Programme (IGBP) land cover classes, at a horizontal resolution of 500 m and annual temporal resolution, from 2001 to present day. It is available from

the Land Processes Distributed Active Archive Center (https://lpdaac.usgs.gov/get_data). We merge land cover classes having similar characteristics into four land types representing the main biomes across the Amazon: tropical forest, savanna, grassland and crops.





## 2.4 Atmospheric conditions

To assess the role of atmospheric conditions on the final elevation of smoke plumes across the Amazon, we analyse data from the second Modern Era Retrospective-analysis for Research and Applications (MERRA-2) reanalysis model simulation (Bosilovich et al., 2015). We focus on the height of the planetary boundary layer (PBL) and the atmospheric stability at the location of our fires. As in previous studies (eg, Kahn et al., 2007; Val Martin et al., 2010), we define the atmospheric stability as the vertical gradient of potential temperature. We use data from MERRA2 at a horizontal resolution of 0.625°longitude by 0.5°latitude, with 42 levels vertical pressure-levels between the surface and 0.01 hPa. MERRA-2 provides hourly PBL height above ground level and potential temperature profiles every 6 hours (0:00, 06:00, 12:00 and 18:00 UT), so we linearly interpolate these data to the time and location of each fire plume origin.

## 2.5 Drought conditions

To determine the presence and magnitude of droughts over the Amazon during our study period, we use the MODIS Drought Severity Index (DSI). The DSI is a global drought index derived by combining the MODIS16 Evapotranspiration (eg, Mu et al., 2007) and the MODIS13 vegetation index (NDVI) data products (Huete et al., 2002). DSI provides drought conditions at global scale for all vegetated areas at 8-day and annual temporal resolutions and 0.5°or 0.05°horizontal spatial resolution for 2000–2011 (Mu et al., 2013). In this work, we use the 8-day temporal resolution DSI and interpolate the data to the time and location of our fire smoke plumes. Following Mu et al. (2013), we further define drought conditions as: "Extreme-Severe" (DSI$\leq -1.2$), "Mild-Moderate" ($-1.2 \leq$DSI$< -0.29$), "Normal" ($-0.29 >$DSI$> 0.29$) and "Wetter than Normal" (DSI$\geq 0.29$).

## 2.6 CALIOP observations

We also use extinction profiles derived from the CALIOP instrument to assess the vertical smoke distribution across the Amazon. CALIOP is a space-borne two-wavelength polarisation lidar (532 and 1064 nm) that flies aboard the CALIPSO satellite (Winker et al., 2013). CALIPSO was launched in 2006 into a sun-synchronous polar orbit of 705 km altitude as a part of the "A-Train" constellation, with an orbit repeat cycle of 16 days. CALIOP collects backscatter and depolarization data that constrain the vertical structure and some properties of aerosols and clouds around the globe (Vaughan et al., 2004; Liu et al., 2009). In addition, CALIOP provides a characterisation of the aerosol type (i.e. dust, polluted dust, marine, clean continental, pollution and biomass burning) based on externally determined surface type along with measured depolarisation ratios, integrated backscatter altitude and colour ratio (Omar et al., 2009). This aerosol-type classification can be used to indicate the sources that probably contribute to aerosol mass loading at specific locations and times where the instrument has coverage.

We use CALIOP Level 2 version 4 day and night data (CAL_LID_ L2_05kmAPro–Standard–V4–10) over the Amazon for the July to November burning season, from 2006–2012. In this work, we filter the data following Ford and Heald (2012). This filtering approach uses cloud-aerosol distinction scores, extinction uncertainty values, atmospheric volume descriptors, extinction quality control flags and total column optical depths, and assumes that extinction observations classified as 'clear





air' have zero aerosol extinction (rather than the fill value). CALIOP daytime retrievals can be biased low due to the noise from scattered solar radiation (Winker et al., 2009; Rogers et al., 2011). However, we analyse both day (i.e., early afternoon, ∼ 13:30 LT equator crossing time) and night profiles to identify any differences in smoke heights, as well as to allow a better comparison with the MISR smoke plumes, which are retrieved during the late morning.

The CALIOP "swath" is ∼100 m wide, so sampling is effectively a curtain. To obtain a climatology of CALIOP smoke plumes across a wide range of conditions as in MISR, we developed an approach to identify individual smoke plumes in the CALIOP data. We first grid all CALIOP aerosol extinction profiles classified as smoke (day and night) at a horizontal resolution of 0.5°x 0.5°over the Amazon region, and a vertical resolution of 250 m, from the surface to 12 km. Within each grid cell, we then determine the vertical distribution of smoke extinction. We define the maximum smoke plume height in each grid cell as

the maximum altitude reached by the extinction classified as smoke. Similar to the MINX definition of median plume height, we consider the median of the CALIOP vertical extinction distribution as the height where most of the smoke is probably concentrated. Smoke does tend to concentrate either in the PBL or in thin layers in the FT (Kahn et al., 2007; Val Martin et al., 2010).

To identify CALIOP smoke plumes associated with active fires, we select only those grid cells that contain at least two

MODIS Collection 6 fire pixels (Giglio et al., 2003), at 80% confidence level or higher, at the time of CALIOP overpass. We also use the mean terrain elevation across each grid cell to reference the maximum and median heights to ground level, as CALIOP provides observations above sea level. Figure 1 shows an example of our approach for the CALIOP observation of September 25th, 2010 at 06:25 UTC. For this example, we identify a CALIOP smoke plume with 1.7 km median and 4.5 km maximum height above ground level. A total of 2460 plumes are characterized with our approach over the Amazon for the

months of July to November, from 2006–2012; about 65% of these plumes are linked to actives fires with some confidence (i.e., having a clear connection to a MODIS fire pixel), and we only consider those in our analysis, i.e., a total of 1600 plumes.

To ensure we do not introduce a bias in the CALIOP plume heights due to the 0.5°x 0.5°horizontal resolution, we also retrieved the smoke plumes for the 2017 burning season at a horizontal resolution of 0.1°x 0.1°, and find no significant differences. For this subset, our 0.5°x 0.5°method returns 131 plumes, with an average altitude of 3.65 km for the maximum

plume heights, whereas the 0.1°x 0.1°method returns 149 plumes, with an average altitude of 3.74 km.

Previous studies have used other CALIOP products to determine the vertical distribution of smoke plumes. The Level 2 Aerosol Layer product is commonly used to analyse smoke plume heights from CALIOP, as it reports the top and base heights of aerosol layers. Tosca et al. (2011) used their smoke layer top altitudes and extinction coefficient profiles over Borneo for September–October 2006. Using the CALIOP Level 1 attenuated backscatter profiles at 532 nm, Amiridis et al. (2010) esti-

mated smoke injection heights from agricultural fires over Europe. They selected only those profiles of constant attenuated backscatter coefficient with height, without strong convection, and that were collocated with MODIS active fire pixels from the Aqua satellite. Recently, Huang et al. (2015) used six years of the CALIOP Level 2 vertical feature mask (VFM) data and aerosol layer products over six regions to investigate the most probable height (MPH) of dust and smoke layers. They used two approaches to obtain MPH: one based on the probability distribution of the vertical profiles of Occurrence Frequency (OF)

(i.e., ratio of number of samples classified as dust or smoke by the VFM to the total samples per grid) and the other as the





probability distribution of the aerosol optical depth (AOD) vertical profiles. So MPH_OF and MPH_AOD correspond to the altitude with the largest OF and mid-visible AOD for a certain type of aerosol. Our definition of CALIOP median plume height is most similar to their MPH_AOD. However, Huang et al. (2015) analysed vertical profiles over large-scale regions (e.g., the entire Amazon or Sahara), whereas in the current work, we analysed and then aggregated the heights for individual smoke plumes.

## 3   Results and discussion

### 3.1   Smoke plume height observations

Figure 2 maps the biomes of the Amazon region for which the MISR plume climatology was developed. Figure 3 presents the time series of the smoke plume heights for the biomass burning seasons (July–November) during the 2005–2012 study

years. We also include a statistical summary of the number of plumes within the time series by year, month, biome and drought conditions in Figure S1 of Supplementary Information (SI). The largest number of plumes is recorded in 2010, with about 25% of the total plumes in the database, whereas the smallest is in 2009 (3%). These two years are the driest and the wettest in the climatology, respectively. Most of the plumes were observed in August and September (85%), at the peak of the burning season in most vegetated location, in the dominant biomes of savanna (48%) and tropical forest (46%), and during dry

conditions (76%). We find an important interannual variability in the type of fires, with dominant fires over tropical forest in 2005 (65%) and 2010 (47%) two of the three drought years in our database as shown in Section 3.4 below, and a majority of savanna fires (54–65%) the rest of the years. We note that a large fraction of the plumes were observed in 2008 (17%) even though it was not a drought year. The majority of the plumes in this record are digitised with blue band retrievals (Table 1), which produce higher quality results in many situations, especially for optically thin plumes over land surfaces.

Throughout the study period, we find significant variability in the smoke plume heights, with altitudes ranging from a few meters (essentially near-surface) to 5 km depending on the biome (Figure 3). Smoke plumes over cropland fires are scarce compared to the other fire types, as these fires are small and tend to be under-detected by MISR (Nelson et al., 2013). We summarise in Table 2 the statistical parameters of the smoke plumes for all observations except the cropland cases. Over the Amazon, the vertical distribution of smoke varies by biome. Statistically, the highest smoke altitudes are detected over

grasslands, with averages for the median and maximum heights of 794 m and 1120 m, respectively, whereas the lowest heights are detected over tropical forest (601 and 845 m, respectively). In all the biomes, more than 85% of the smoke is located at altitudes below 2 km (Fig S2 in SI).

Similar altitudes and distributions have been found across comparable fires in other parts of the world. For example, altitudes on the range of 700–750 m were detected over the tropical forest in central America and Indonesia (Val Martin et al., 2010;

Tosca et al., 2011). In contrast, smoke plume heights over the Amazon are substantially lower than smoke plumes observed over the boreal biomes (960–1040 m) (Kahn et al., 2008; Val Martin et al., 2010). There are several factors that influence smoke altitudes and contribute to the differences between biomes, e.g., fire intensity, availability of fuel, combustion efficiency,





atmospheric stability, and entrainment (eg, Lavoué et al., 2000; Trentmann et al., 2006; Luderer et al., 2006; Kahn et al., 2007, 2008; Val Martin et al., 2012). We assess some of these factors next.

## 3.2 Effect of atmospheric and fire conditions on smoke plumes

We explore the relationship between smoke plume height, fire characteristics (i.e., MODIS FRP and AOD) and atmospheric conditions derived in the vicinity of the fires throughout the burning season, across the major biomes in the Amazon except cropland. For atmospheric conditions, we focus both on how smoke plume height relates to boundary layer height and on the effect of atmospheric stability on plume rise. We consider atmospheric stability conditions above our fires as the average of the atmospheric stability over the atmospheric column (K/km; Section 2.4) from the surface, at the origin of the fire, to the maximum altitude that smoke reached in the atmosphere. We add a buffer of 10% to the maximum altitude to account for any potential influence that the atmosphere above the plume might have over the column. We include in Table 2 a summary these main parameters.

Consistent with previous studies (e.g., Val Martin et al., 2010, 2012; Sofiev et al., 2009; Amiridis et al., 2010), we find that the highest-altitude smoke plumes tend to be associated with highest MODIS FRP values, though there is significant variability in the relationship in all the biomes ($r^2$=0.2; Figure S3 in SI). Smoke plumes detected over tropical forest fires have the lowest FRP (209 MW) and largest AOD values (0.51) on average (Table 2). The other two main biomes (savanna and grassland) show similar FRP and AOD values (360–421 MW and 0.33–0.35, respectively). Tropical forest has deeper root systems, which allows fires to access deeper soil layers (Nepstad et al., 2008) that can maintain higher moisture content and lower oxygen availability than other biomes, such as grasslands. High moisture content and low oxygen in fuels favour smouldering rather than flaming fires, which in turn tends to produce greater smoke emission but lower radiant emissivity (Kauffman et al., 1995). Therefore, the low FRP and high AOD in tropical forest fires are consistent with these conditions, in which smouldering fires predominate, whereas high FRP and low AOD are typical from dryer, less dense fuels, e.g. savanna and grassland, which tend to produce flaming fires (Giglio et al., 2006). In addition, high smoke opacity and tree canopy obscuring the fire-emitted 4-micron radiance as viewed by MODIS, as well as low radiant emissivity, rather than just low fire intensity, probably contribute to these differences (Kahn et al., 2008).

The atmospheric stability structure affects the vertical motion of smoke and is a key factor in plume rise, either enhancing or suppressing the lifting. Some studies have shown the important role that atmospheric stability plays in plume rise (eg, Kahn et al., 2007, 2008; Val Martin et al., 2010; Amiridis et al., 2010). For instance, Val Martin et al. (2012) showed that, for fires in North America, smoke plumes that inject to high altitudes tend to be associated with higher FRP and weaker atmospheric stability conditions than those plumes at low altitudes, which tend to be trapped within the boundary layer. Similar results were found for agricultural fires over eastern Europe (Amiridis et al., 2010).

To analyse the influence of atmospheric stability over Amazon fires we divide our plume dataset into two groups having weak and strong atmospheric stability conditions based on MERRA-2 reanalysis. Over the Amazon, and at the locations and times studied, atmospheric stability ranges from −3 to 23 K/km. We designate atmospheric stability < 2 K/km as 'weak', and atmospheric stability > 4 K/km as 'strong'. Each group contains about 30% of plumes in the database. Figure 4 shows the




vertical distribution of smoke stereo-height retrievals for the plumes classified under weak and strong atmospheric stability conditions. Our comparison supports previous observations that plumes under weak atmospheric conditions tend to inject smoke to higher altitudes than those experiencing strong stability, with average maximum plume heights of 1150 m and 654 m, respectively. A similar pattern is found for the average of the median plume heights (821 and 482 m, respectively). Weak

atmospheric stability conditions are also associated with deeper PBLs (∼1500 m) than strong stability conditions (∼1200 m) (not shown).

Atmospheric conditions also depend on biome type. We find that tropical forest fires tend to be associated with more stable atmospheric conditions than grassland fires (4.2 versus 2.5 K/km). A narrower PBL is also observed over tropical forest (1270 m) compared to grassland (1620 m). Tropical forests typically have higher relative humidity conditions and more con-

stant temperatures than grasslands, which favours more stable conditions and lower PBL heights (Fisch et al., 2004). We note that our dataset is all acquired at Terra overpass time, which occurs between about 10–11 am LT. This might produce a bias toward the more stable atmospheric conditions that occur preferentially during the morning; later in the afternoon convection tends to become more important (Itterly et al., 2016).

The seasonality of these parameters in combination with the fire intensity determine the vertical smoke plume rise over the

Amazon and the ability of these fires to inject smoke to high altitudes and/or into the FT.

### 3.3   Seasonality of smoke plumes heights

Figure 5 shows the seasonal cycle of maximum plume height with FRP, AOD, and atmospheric conditions over the major Amazon biomes. We further disaggregate these observations by biome, season and dry/wet years in Table S1 in SI. For these biomes, we find minimum plume heights of 600–750 m in July and maximum plume heights of 900–1400 m in October and

November. Similarly, over tropical forest and grasslands, MODIS FRP values follow the plume-height patterns, with maximum values toward the end of the burning season (180–200 MW), compared to the early season (90 MW). For savanna fires, MODIS FRP remains mostly constant throughout the season (∼150-200 MW). Savannas are known to be fire-adapted, and combustion efficiency typically remains constant throughout the season (Van der Werf et al., 2010). All these patterns are similar in wet and dry years, although larger MODIS FRP values are observed over savanna and grassland fires in dry years (Table S1).

Some previous studies show the seasonal peak in MODIS FRP over the Amazon earlier, in August–September (Tang and Arellano, 2017). However, their work relies on the maximum MODIS FRP detected by the Terra and Aqua satellites (four times/day) over the Amazon, whereas our seasonality shows the average MODIS FRP observed by Terra, collocated with the MISR smoke plume observations (once/day). In addition, MISR path is substantially narrower than MODIS (380 versus 2330 km), and many fires detected by MODIS are not observed by MISR. Our seasonality thus captures the fire intensity that drives

the smoke plumes detected specifically by MISR, i.e., only at about 10:30 AM local time, and the seasonal differences provide at least some indication of possible bias introduced by the MISR sampling of fires.

In contrast to the seasonality of plume heights and fire intensity, the peak monthly AOD occurs in September across the major biomes, with median AOD of 0.6 in tropical forest and 0.3 in savanna and grasslands, compared to AOD values of 0.04–0.1 in July and November. Over the Amazon, total AOD correlates well with the number of fires, and both tend to peak



during September each year (Mishra et al., 2015). Baars et al. (2012) reported optical depths in the polluted biomass burning season (July–November) six times larger (on average) than in the pristine wet season (December–June), with highest values in September and October. In our dataset, September, together with August, are the months when the largest number of plumes are detected (Figure S1 in SI). However, our monthly statistics might be influence by the number of observations in each

month. For example, the number of fires in August is driven by year 2010, in which an unusually large number of fires are observed, compared to the other August months. In addition, the large monthly median values in November are based on the fewest number of plumes (Figure S1 in SI), although the few fires detected by MISR for those months were large and intense.

Boundary layer heights and atmospheric stability conditions may also vary by biome and throughout the season, influencing plume-rise spatial and temporal distributions. On a seasonal basis, the PBL height does not follow a clear cycle in any of our

biomes, but higher PBL heights are observed over grassland fires (Table 2) and across all the biomes during dry years (Table S1). More stable atmospheric conditions are found at the beginning (3.6 K/km in July) compared to the end of the burning season (1.9 K/km in November).

Previous studies have shown that a substantial fraction of smoke is injected above the boundary layer (i.e., into the FT), although this fraction varies depending on biome and fire type. For tropical fires over central America and Indonesia, smoke

from about 4–6 % of fires is reported to reach the FT (Val Martin et al., 2010; Tosca et al., 2011). This fraction is larger for boreal fires (>16%), where fires are more intense and the BL is typically lower than in tropical regions (Val Martin et al., 2010; Kahn et al., 2008; Val Martin and Kahn, 2018). Following these studies, we consider that smoke reaches the FT when the median height of the plume is at least 500 m above the PBL height. This is a conservative definition that takes into account uncertainties in MINX and MERRA (eg, Kahn et al., 2008; Val Martin et al., 2010; Tosca et al., 2011). Because fires over the

Amazon tend to be smaller in size than those in boreal forests, we also consider a less conservative definition. We assume a plume is injected into the FT when the maximum plume height is at least 250 m above the PBL height. We understand that this is an upper limit, but it provides a bracket to our results. We include in Table 2 the percentage of the smoke plumes injected into the FT for both definitions, and present in Figure 6 the seasonality of these percentages. Our analysis shows that fires at the end of the burning season are more likely to inject smoke in the FT, with 15–40% in November versus 2–10% in July, and 5–22% at

the peak of the burning season (August–September). This pattern seems to be related to a combination of more intense fires and less stable atmospheric conditions. We find no influence of the monthly PBL depth variability, although deeper PBL heights are found across the Amazon in drier conditions (i.e., over grassland fires and/or dry years). Interestingly, our analysis also shows a slightly larger percentage of fires inject smoke into the FT over grassland (5–19%) compared to tropical forest (3–15%). As mentioned above, grassland fires are associated with high PBL heights, but also with large FRP values, suggesting that these

fires are energetic enough to produce the buoyancy needed for the smoke to reach the FT.

## 3.4   Interannual variability of smoke plumes and drought conditions

We use MODIS DSI to assess the effect of drought conditions on smoke plume rise and the extent that these conditions control the interannual variability of smoke plumes across the region. We present the interannual variability of MISR plume heights, MODIS FRP and MISR AOD in Figure 7, and summarise the annual averages of MODIS DSI, atmospheric stability, PBL



heights and percentage of smoke plumes in the FT in Table 3. In addition, we include the annual relationship of MISR plume heights, MODIS FRP and MISR AOD with MODIS DSI, and the percentage of plumes in the FT per drought level in Figure 8. In our dataset, 76% of plumes are recorded under extreme-mild drought conditions versus 7% plumes in wet conditions, as discussed above. During drought years (2005, 2007 and 2010), smoke plumes register the lowest averaged MODIS DSI values

(-0.89, -0.91 and -1.50, respectively), compared to the other years in the climatology (-0.63–0.18). Note that DSI is higher in wetter years.

We find a significant positive relationship between MISR maximum plume heights and MODIS DSI (r=0.7) in tropical forest and savanna fires, with higher maximum plume heights in wet (1000–1100 m) than severe drought conditions (800–900 m) (Figure 8). Consistently, on an annual basis, these two biomes show the lowest smoke plume heights during dry years (Figure

S4 in SI). Smoke plume heights in grassland fires, however, do not show any strong relationship with DSI (r=0.1) or a clear interannual variability driven by droughts (Figure S4). In general, lowest median smoke heights are observed in our dataset during the drought years of 2005 and 2010 (Figure 7), which are driven by tropical forest observations as they are the dominant biomes (Figure S1).

The relationship between MODIS FRP and drought levels over the Amazon is not straightforward on an annual basis as we

do not observe any clear interannual variability on FRP driven by droughts in Figure 7. However, our analysis shows some patterns when we subdivide the data by biome (Figures 8 and S4 in SI). For example, we find a significant positive relationship between MODIS FRP and DSI (r=0.6) in tropical forest, with lower FRP in extreme dry than normal-wet conditions (170 versus ∼250 MW; Figure 8). Contrariwise, savanna and grassland fires have higher FRP in extreme and mild dry than wet conditions (∼500 MW versus 250 MW), although their relationship is weak (r=-0.4). As mentioned above, interpretation of

FRP can be complicated by factors such as overlying smoke opacity and fire emissivity (Kahn et al., 2008).

The relationship between smoke plume heights, FRP and drought conditions over the Amazon is somewhat complex. Drought conditions over the Amazon increase fuel flammability and number of fires, but not necessarily increase smoke elevation. Drought also decreases fuel load, i.e., fuel available to burn, specially over grassland. Tang and Arellano (2017) reported that drought in the Amazon favours understory fires for tropical forest, which are dominated by smouldering combustion and

are linked to low altitude smoke plumes. In addition, spatial changes in drought location may influence the type of biome affected and hence the type of fire regime in a given year. For example, drought in 2005 was located at the northeastern and central regions, and the large majority of the plumes recorded by MISR (65%; Figure S1) are from tropical forest fires, i.e., related to smouldering and fires that inject smoke to lower altitudes. In 2007, drought shifted to the southeastern region, and the majority of the plumes (60%; Figure S1) are from savanna and grassland fires associated with more flaming burning conditions,

i.e., higher FRP and smoke plume altitudes. Our analysis supports this observation. In 2005, a drought year, smallest MODIS FRP (150 MW) and lowest smoke plume heights (750 m) are recorded over tropical forest (Figure 8), whereas in 2007, another drought year, larger FRP (500 and 750 MW), associated with higher smoke plume heights (1100 and 1300 m), are recorded over savanna and grassland fires, respectively.

In addition to the influence of droughts in controlling the type of fires, droughts can also influence atmospheric conditions.

We find that during drought years, PBL heights tend to be about 200 m deeper than in wet years (Table 3). On annual basis,





atmospheric stability does not vary significantly, with values of ∼3–4 K/km, across the Amazon for the averaged biomass burning season (Table 3). We also observe that a lower percentage of fires inject smoke plumes into the FT in drought compared to non-drought years (2–18% versus 4–28%; Table 3). On a biome basis, tropical forest fires inject a larger percentage of smoke plumes into the FT in wet than extreme-dry conditions (12 versus 20%, Figure 8), and swallower PBL heights may partially

explain the larger percentage of MISR plumes detected in the FT during non-drought years. Contrariwise, grassland fires, although with fewer observations, inject more smoke plumes into the FT during extreme dry than wet conditions (25% versus 13%, Figure 8). These fires are associated with high FRP values in dry conditions and this extra fire energy may be enough to produce the buoyancy needed to lift smoke directly into the FT, regardless of the PBL height. Note that for this analysis we subdivide the data per MODIS DSI and biome, regardless of the year.

Consistent with previous studies that have shown significant positive relationships between drought conditions and aerosol loading (e.g., Reddington et al., 2015; Tang and Arellano, 2017), we find a significant relationship between MISR AOD and MODIS DSI on an annual basis in tropical forest and savanna fires (r=−0.7 and p< 0.01; Figure 8). Years with drier conditions have almost a factor of three greater AOD compared with years with wet conditions. In addition, MISR AOD shows significant interannual variability, with largest AOD values recorded in 2005, 2007 and 2010 (0.4–0.6; Figure 7), and in particular over

tropical forest fires (0.6, Figure S4 in SI). Our results suggest that fires during drought periods might significantly degrade regional air quality, as they are associated with low smoke altitude and large aerosol loading.

### 3.5   CALIOP smoke plume observations

To further investigate smoke rise over the Amazon, we develop a climatology of smoke plume heights using CALIOP extinction profiles (section 2.6). We identify a total of 1600 CALIOP smoke plumes linked to active fires from July–November, 2006–

2012 (Fig S5 in SI). Although the CALIOP climatology is 1/3 in size of the MISR climatology, these datasets agree well with respect to the temporal and spatial distributions. Similar to MISR, the largest number of plumes corresponds to years 2007 and 2010 (22 and 29%), whereas the lowest records are in 2009 and 2011 (4 and 7%). Most of the CALIOP plumes are also recorded at the peak of the biomass burning season (September; 51%) and over savanna and tropical forest (37 and 57%, respectively).

Figure 9 displays the time series of derived median and maximum heights, for day and night-time observations. We include both daytime and night-time CALIOP observations to assess any day-night differences in smoke plume rise. Similar to the MISR climatology, we find large variability in the CALIOP smoke plume heights; the median heights range from 0.8–4.4 km (daytime) and 1.1–4.5 km (night-time). Maximum smoke plume heights are obviously higher, typically spanning 1.8–5 km (daytime) and 2.4–5.8 km (night-time). About 18 maximum plume height observations fell above 6 km (shown saturated at

6 km in Figure 9). Here we examine the vertical distribution of aerosol plumes individually. Ten cases show high altitude smoke (> 6 km) in a layer that extends through the column to near-surface (Figure S6 in SI, left panel), implying that smoke from the active fire below was lifted by fire-induced buoyancy, atmospheric processes, and/or both. The remaining cases show that high-altitude smoke was disconnected from the surface smoke layer (Figure S6, right panel), and we suggest that this smoke could be residual smoke from older fires, smoke transported from the source and concentrated in an elevated layer, aerosol that



was wrongly classified as smoke by the CALIOP algorithm, and/or the result of CALIOP not being able to detect lower-level aerosol due to smoke thick aloft or the presence of clouds in the column. We chose to include these observations in our analysis as these cases only represent 1% of the total observations within the climatology and do not impact the overall results shown here.

Our initial objective was to compare the CALIOP with the MISR plumes to assess the diurnal smoke evolution, as CALIOP has a later sampling time than MISR over the Amazon.(14:00–15:00 LT versus 10:00–11:00 LT). However, despite our effort to develop a comprehensive CALIOP climatology none of the CALIOP plumes coincide with the MISR plumes. As previous studies discuss (eg, Kahn et al., 2008; Tosca et al., 2011), CALIOP and MISR, in addition to having different sampling times, also have different swath widths (380 km versus 70 m). These differences make it difficult to observe the same fire on the same

day, but they make CALIOP and MISR observations complementary (Kahn et al., 2008).

     Figure 10 summarises the median and maximum heights for day and night-time smoke plumes per biome, season and wet/dry years. We also include the MISR plume heights for comparison. Average daytime median plume heights range from 2 km (tropical forest and savanna) to 2.3 km (grassland), and night-time median plume heights range from 2.2 km (tropical forest) to 2.4 km (grassland). Maximum plume heights are similar across all biomes (∼3.2 km). Similar to MISR, CALIOP

detects higher smoke plumes during the late burning season (2.1 and 3.3 km, for the average of median and maximum plume heights, respectively) than the early season (1.9 and 3.0 km). In contrast, CALIOP observes smoke at higher altitudes during dry (2.2 and 3.3 km) than wet years (2.0 and 3.1 km). As discussed above at the time/location of MISR observations, a deeper PBL is observed in dry compared to wet years. The PBL height is expected to grow further by the time of the CALIOP observation later in the day, thus a deeper PBL during drought conditions may partially explain higher altitudes observed by

CALIOP under drier conditions.

     Night-time plume heights are on average ∼250 m higher than daytime plume heights. Differences between day and night-time CALIOP observations have been attributed in the past to a low bias in the daytime retrievals due to noise from saturation of solar radiation (eg Winker et al., 2009; Huang et al., 2015). Therefore, our difference in day and night-time CALIOP plume heights might result from differences in data quality rather than reflecting smoke diurnal variability.

Similar CALIOP smoke plume height values over the Amazon were reported by Huang et al. (2015). Using the CALIOP vertical feature mask and AOD profiles, they reported an average for the most probable smoke height of 1.6–2.5 km for September fires. Their definition is comparable to our CALIOP median plume height, which produced a value of 2.3±0.7 km for the September months. Other CALIOP smoke plume heights have been reported over eastern Europe (1.7–6 km) and several regions and biomes across Asia (0.8–5.3 km) (Amiridis et al., 2010; Labonne et al., 2007; Tosca et al., 2011; Vadrevu et al.,

30    2015).

     In our study, CALIOP observes smoke at systematically higher altitudes than MISR, with median plume heights up to 1.4 km higher (2.2 km for the maximum plume heights). However, CALIOP still shows that the majority of the smoke is located at altitudes below 2.5 km above ground. Tosca et al. (2011) found similar differences between CALIOP and MISR (1–2.8 km) in peatland fires over southeastern Asia. CALIOP height retrievals are more sensitive to thin aerosol layers than MISR stereo

analysis, so CALIOP is more likely to detect low-density smoke at plume-top (Kahn et al., 2008), e.g., smoke that has been



lifted later during the day by convection, air mass advection or fire buoyancy (Kahn et al., 2008; Tosca et al., 2011). Although we only select CALIOP plumes that are directly linked to active fires with some confidence, fires can burn for several days (and even weeks); in particular, deforestation fires can leave residual smoke layers over the region. Therefore, our CALIOP plume heights may include low-density smoke at higher altitudes, possibly from old fires.

Some previous studies with MISR smoke plume height have also analysed the altitude of 'smoke clouds', that is, dispersed smoke not easily associated with a particular fire (Val Martin et al., 2010; Tosca et al., 2011). Smoke clouds tend to occur at higher altitudes than smoke plumes; they tend to represent fire plumes at a later stage of evolution. Over Borneo peatland fires, Tosca et al. (2011) show that MISR smoke clouds and CALIOP smoke plumes had similar altitudes for their period of study. The analysis of smoke clouds over the Amazon may support the expectation that the plume heights tend to grow even larger than observed by MISR later in the afternoon. In addition, transported smoke is more likely to have stayed aloft longer than near-source smoke, and would therefore have more opportunity to mix upward.

## 4 Summary and conclusions

A climatology of smoke plumes from MISR and CALIOP observations is used to characterise the magnitude and variability of smoke altitude across the Amazon during eight biomass burning seasons. Biome type, fire and smoke properties (FRP and AOD), atmospheric conditions (PBL and atmospheric stability) and regional drought state are included in the analysis, to explore the degree to which each contributes to the observed variability.

Analysis of the smoke plume climatology shows large differences in smoke-plume elevation over the main biomes in the Amazon, with heights ranging a few hundred meters to 5.2 km above ground level. Smoke from plumes observed by MISR (10:00-11:00 LT) is mainly concentrated at altitudes below 1.5 km. As expected, smoke plume elevations are higher in our CALIOP climatology, ranging from 0.8 to 6 km at daytime (14:00-15:00 LT ), although the majority are concentrated below 2.5 km. We find that CALIOP smoke plume heights are about 1.4–2.2 km higher than MISR smoke plumes, most likely due to greater sensitivity to very thin aerosol layers (Kahn et al., 2008; Flower and Kahn, 2017). Thus, our CALIOP plume climatology includes fresh smoke from active fires and low-density smoke at higher altitudes, some of which might be from old fires. Our results show that over the Amazon, and similar to other fire regions studied previously, on average, smoke- plume heights tend to increase later in the afternoon due to greater near-surface convection, greater fire intensity, and possibly self-lofting. Direct injection of smoke at altitudes higher than 6 km (middle to upper troposphere) did not seem to be significant over the Amazon during our study period.

For our main biomes in the Amazon, smoke plume heights are substantially lower over moist tropical forest fires (0.8 km, maximum plume height definition) than grassland fires (1.1 km), although grassland smoke fire plumes represent a small fraction (4%) of cases in the climatology. The MISR and CALIOP Amazon plume climatologies show a well-defined plume height seasonal cycle in the main biomes, with larger heights toward the end of the burning season. Using MODIS FRP and MERRA-2-estimated atmospheric stability conditions, we determine that higher smoke-plume elevations in October–November are the result of the combination of more intense fires and a less stable atmosphere. Less than 5% of the fires inject



smoke into the FT (i.e., Median Plume–PBL height> 500 m) using a conservative criterion, although an additional 15–19% of the fires may inject some smoke based on a looser criterion (i.e., Maximum Plume–PBL height > 250 m). This fraction increases throughout the season, with about 15–40% of the fires injecting smoke above the FT in November.

Previous studies have shown a direct connection between droughts, large-scale climate process (e.g., ENSO) and the number of fire occurrences (eg Alencar et al., 2006; Inness et al., 2015). We find a negative relationship between MISR plume heights and drought conditions in tropical forest fires, as wet years show smoke plume altitudes 300 m higher than dry years. Tang and Arellano (2017) reported that drought conditions over the Amazon favour understory fires, for which smouldering combustion dominates, favouring lower smoke injection heights. In addition to low-altitude smoke, we find that drought conditions are also related to deeper PBL heights, which can reduce the frequency with which smoke is able to reach the FT.

A relationship between fire intensity (FRP) and drought conditions is not clear in our study. We detect highest FRP values in grassland fires during dry periods, and lowest FRP values for tropical forest fires under similar dry conditions, but without a significant relationship between FRP and DSI, nor any interannual variability of FRP driven by droughts. This lack of relationship may be due to the different locations of drought in different years, the type of fires recorded by MISR in a given year, and/or the low performance of MODIS FRP under dense smoke conditions.

Consistent with previous observations, we find larger MISR AOD during drought compared to non-drought periods. Our analysis confirms the important effect that biomass burning has on smoke aerosol loading over the region, from the surface to the lower free troposphere. Strong land management policies to control fires over the Amazon may become crucial as increases in drought frequency are projected in a future climate (Malhi et al., 2008); this would have important consequences for fire activity and thus air quality.

Observations from both CALIOP and MISR provide a way to study smoke plume heights across the Amazon during the biomass burning season. Ultimately, this information will help improve the representation of biomass burning emissions in Earth system atmospheric models, and should aid our understanding of the feedbacks between droughts, terrestrial ecosystems and atmospheric composition over the region. A next step in our work includes the evaluation of the influence of smoke plume height on the atmospheric composition over the southern hemisphere, based on insights from the analysis of the smoke plume climatology across the Amazon, and further application of this approach to other geographic regions.

*Acknowledgements.* This work is supported by the NASA Award Number NNX14AN47G. LGA PhD program was supported by the UK EPRSC and The University of Sheffield Chemical and Biological Engineering Department. MVM was partially supported by the Leverhulme Trust through a Leverhulme Research Centre Award (RC-2015-029) The work of RAK is supported in part by NASA Climate and Radiation Research and Analysis Program under Hal Maring, NASA Atmospheric Composition Program under Richard Eckman, and the NASA Earth Observing System MISR project. We thank the University of Sheffield OnCampus Placement Program for funding two summer students who contributed to the digitising effort.





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



**Table 1.** Summary of MISR smoke plumes over the Amazon domain (2005–2012)

| | Number of Plumes[a] | | | | |
| Year | Total | Blue Band | Red Band | MINX version | Reference |
|---|---|---|---|---|---|
| 2005 | 927 | 122 | 805 | v3/v4 | This study |
| 2006 | 513 | 501 | 12 | v2/v4 | MPHP[b]/This study |
| 2007 | 858 | 670 | 188 | v2/v4 | MPHP[b]/This study |
| 2008 | 889 | 889 | 0 | v3.1 | MPHP2[c] |
| 2009 | 150 | 55 | 95 | v3/v4 | This study |
| 2010 | 1373 | 0 | 1373 | v3 | This study |
| 2011 | 320 | 320 | 0 | v4 | This study |
| 2012 | 363 | 30 | 333 | v3/v4 | This study |

[a] Total number of plumes, and number of plumes digitised with blue/red band retrievals

[b] MISR Plume Height Project; data from https://misr.jpl.nasa.gov/getData/accessData/MisrMinxPlumes/

[c] MISR Plume Height Project2; data from https://misr.jpl.nasa.gov/getData/accessData/MisrMinxPlumes2/

**Table 2.** Statistical summary for main smoke plume parameters and atmospheric conditions[a].

| | Tropical Forest | Savanna | Grassland |
|---|---|---|---|
| Median Height (m) | 601 ± 339 | 743 ± 422 | 794 ± 471 |
| Max Height (m) | 845 ± 499 | 1040 ± 585 | 1120 ± 653 |
| MODIS FRP (MW) | 209 ± 537 | 360 ± 658 | 421 ± 614 |
| AOD (unitless) | 0.51 ± 0.34 | 0.33 ± 0.28 | 0.35 ± 0.29 |
| Atm Stab (K/km) | 4.21 ± 2.97 | 3.16 ± 3.16 | 2.52 ± 2.50 |
| BL Height (m) | 1270 ± 514 | 1490 ± 507 | 1620 ± 530 |
| Plumes in FT (%)[b] | 3–15 | 4–17 | 5–19 |
| Number | 1744 | 2084 | 166 |

[a] Reported the average±SD and number of observations

[a] Reported range from more and less conservative definition of plume in the FT (see text for explanation).

*Competing interests.* The authors declare no competing interests that could have influenced the interpretation of this work





**Table 3.** Summary of the main atmospheric parameters calculated at the location of the plumes per year[a].

| Year | Number | BL height (m) | Atm. Stab (K/km) | % in FT[c] |
|---|---|---|---|---|
| 2005[b] | 927 | 1370 ± 546 | 4.32 ± 3.01 | 3–13 |
| 2006 | 513 | 1210 ± 518 | 3.50 ± 2.89 | 6–25 |
| 2007[b] | 858 | 1380 ± 539 | 3.96 ± 3.30 | 3–18 |
| 2008 | 889 | 1480 ± 558 | 3.02 ± 2.28 | 4–23 |
| 2009 | 150 | 1100 ± 377 | 3.22 ± 2.60 | 4–27 |
| 2010[b] | 1373 | 1550 ± 498 | 3.69 ± 3.53 | 2–7 |
| 2011 | 320 | 1150 ± 296 | 2.73 ± 2.38 | 8–28 |
| 2012 | 363 | 1330 ± 453 | 3.20 ± 3.29 | 4–13 |

[a] Reported the average±SD

[b] Drought years

[c] Reported as percentage of plumes where [Median Plume–BL Height]> 0.5 km-[Maximum Plume–BL Height]> 0.25 km (see text for explanation)

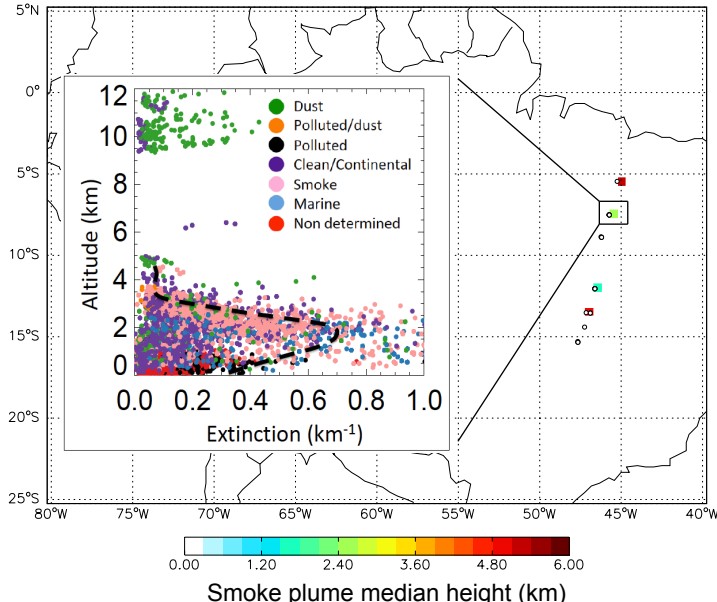

**Figure 1.** Example for the CALIOP smoke plume characterisation. Estimated smoke plume median heights (0.5x0.5) for September 25th 2010 (nighttime) overpass is shown in map. MODIS active fire pixels associated with the CALIOP plumes are represented with open circles on the map. The insert displays the vertical distribution of extinction with values coloured by classified aerosol types. Dashed black line represents the averaged extinction profile for the aerosols classified as smoke (pink dots).




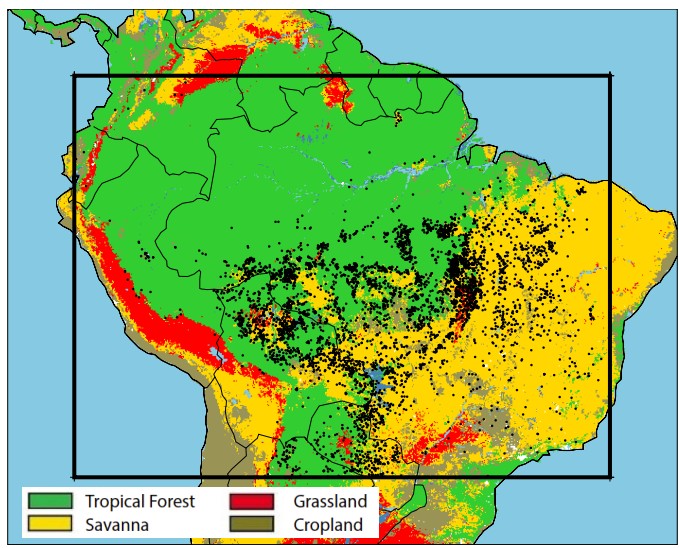

**Figure 2.** Locations of the MISR plumes analysed (black dots) over the four main biomes considered in the study. The black square represents the Amazon domain.

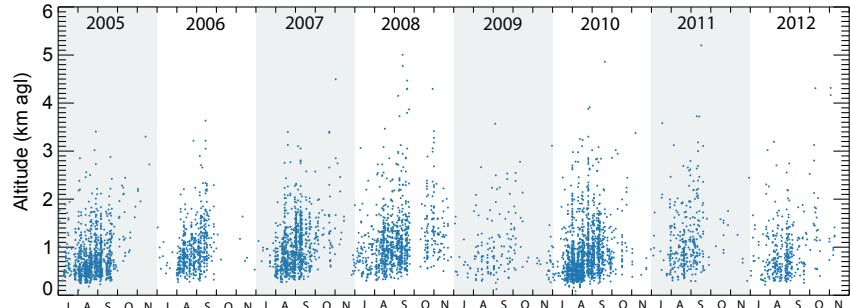

**Figure 3.** Time series of the 2005–2012 MISR Amazon smoke-plume-height climatology, covering the July-November burning season for each year. Each blue dot represents the maximum smoke height above ground level (agl) for one plume.





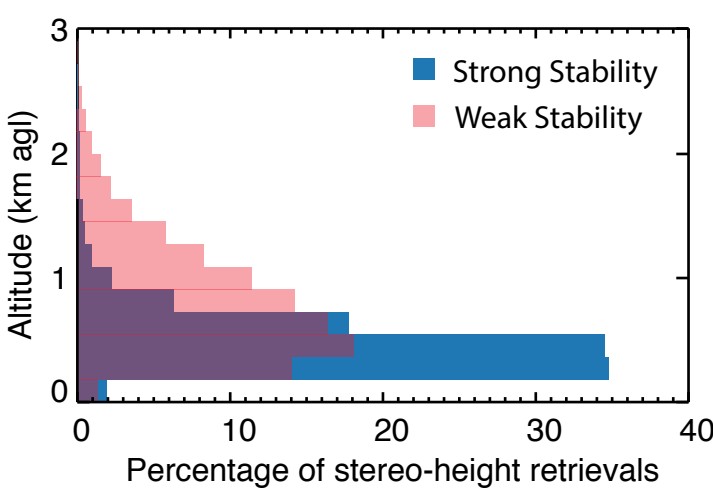

**Figure 4.** Vertical distribution of MISR stereo-height retrievals for all the plumes analysed, under strong (blue) and weak (red) atmospheric stability conditions.





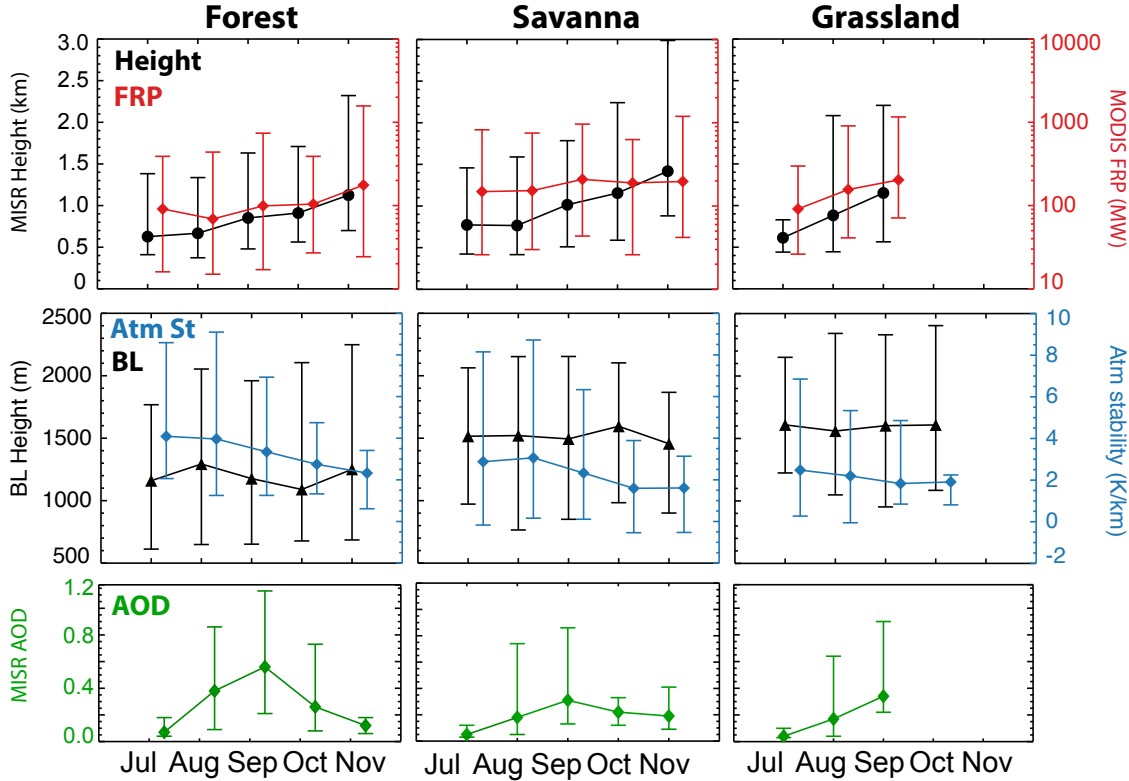

**Figure 5.** Seasonal cycle of MISR smoke plume maximum height above the terrain (black circles), MODIS FRP (red diamonds), PBL heights (black triangles), atmospheric stability (blue diamonds) and MISR AOD (green diamonds). Monthly median values are shown for tropical forest, savanna and grassland biomes. Vertical bars indicate the 10th and 90th percentile. Distributions with fewer than 10 observations are omitted and all years are included

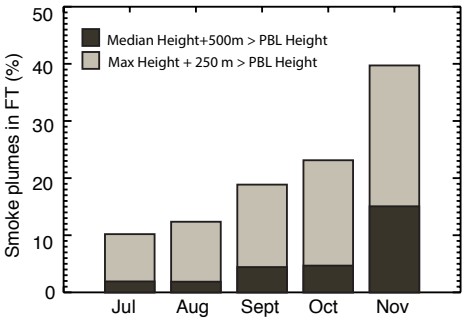

**Figure 6.** Seasonal variation of Amazon plume injection above the PBL (percent). Bar plots indicate the average of [Median Plume–PBL Height]> 0.5 km (dark grey) and [Maximum Plume–PBL Height]> 0.25 km (light grey) (see text for explanation).



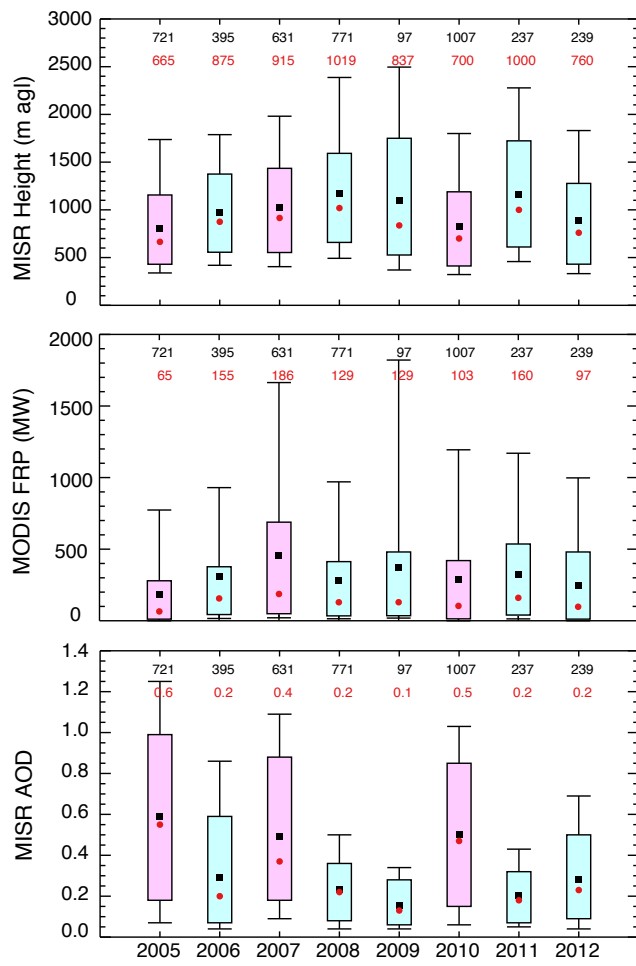

**Figure 7.** Interannual variability of MISR maximum plume heights above the terrain, MODIS FRP and MISR AOD, for the aggregate of tropical forest, savanna and grassland. Bar plots indicate the distribution of the data for each year. The medians (red circles) and the means (black squares) are shown along with the central 67% (box) and the central 90% (thin black whiskers). The number of observations (in black) and the median values (in red) included in each distribution are given at the top of the plot. Drought years are in pink and non-drought years in light blue. The same data, stratified by biome type, are plotted in Figure S4 in SI.




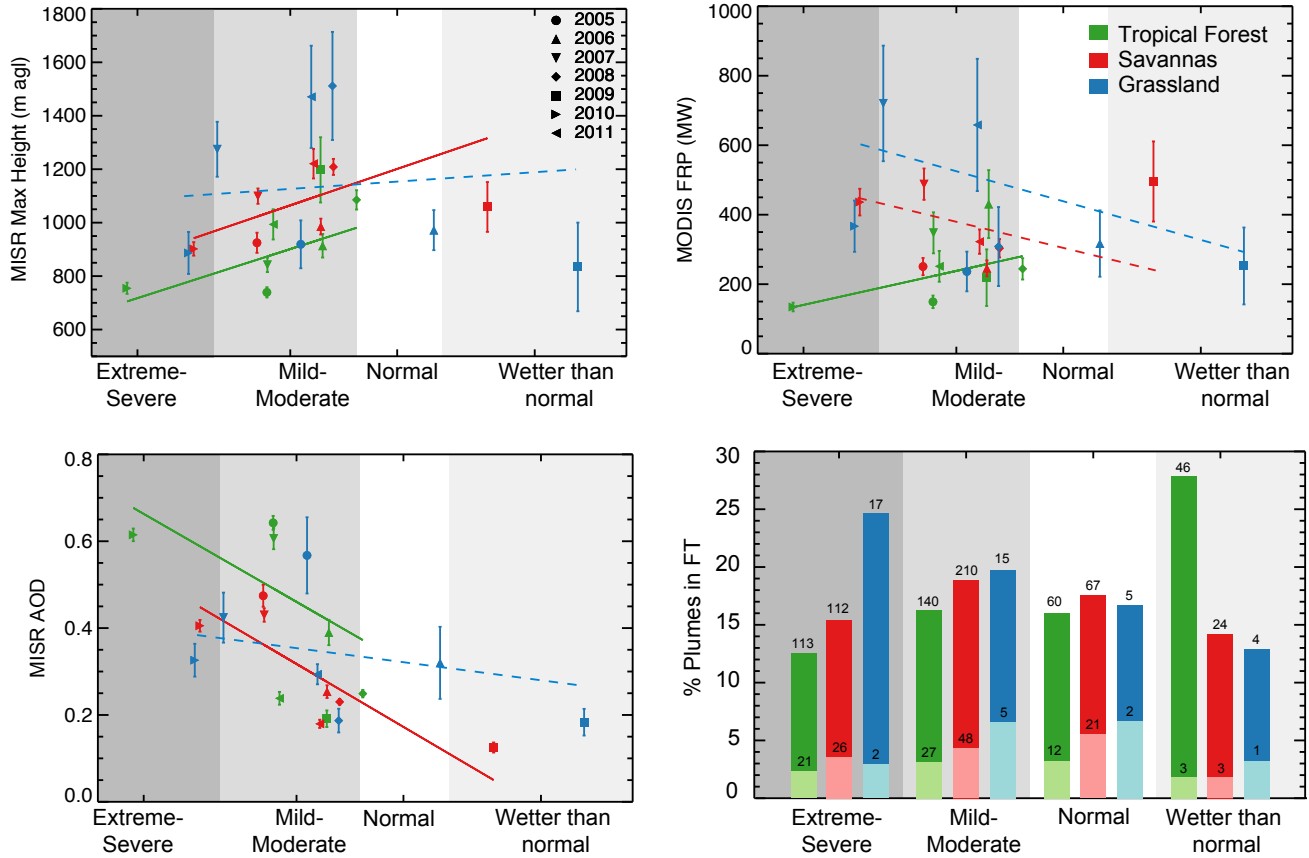

**Figure 8.** Relationship between annual MODIS DSI at the location of the plumes and MISR maximum plume height, MODIS FRP and MISR AOD for tropical forest (green), savanna (blue) and grassland (red). Regression lines are weighted by the number of plumes in each year; relationships with absolute $r < 0.4$ are plotted in dashed lines. Also included percentage of smoke plumes in the FT in each biome and by drought condition. Bar plots indicate the average of [Median Plume–PBL Height]$> 0.5$ km (light colour) and [Maximum Plume–PBL Height]$> 0.25$ km (dark colour), based on MERRA-2 PBL heights (see text for explanation).




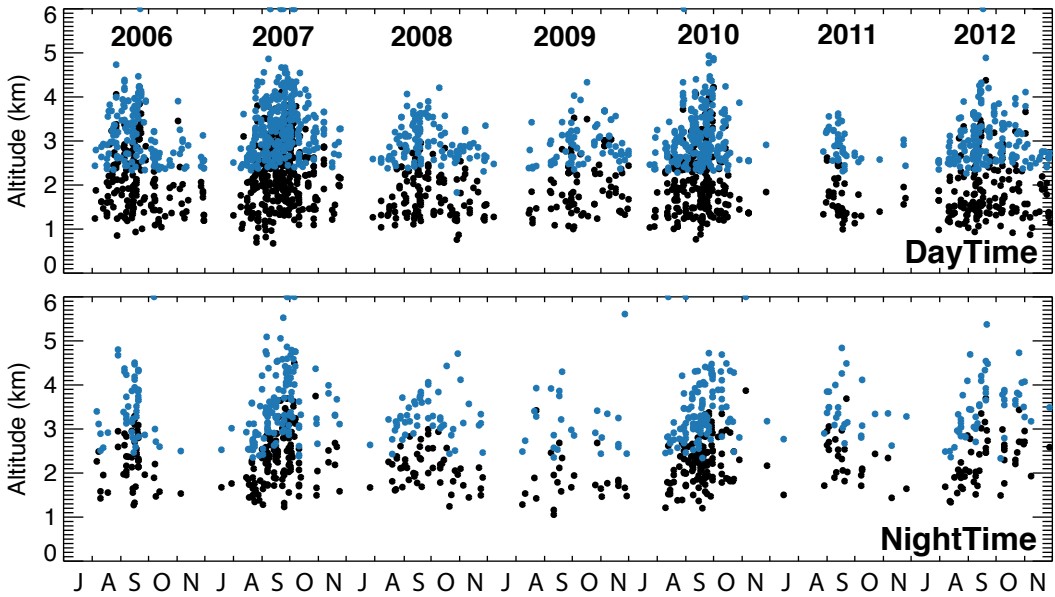

**Figure 9.** Time series of the CALIOP smoke plumes (2006–2012) for daytime and nighttime observations. Each dot represents the maximum (blue) and median (black) smoke plume height above the terrain. Eighteen points for which the CALIOP height exceeds 6 km are plotted at the top of the charts.

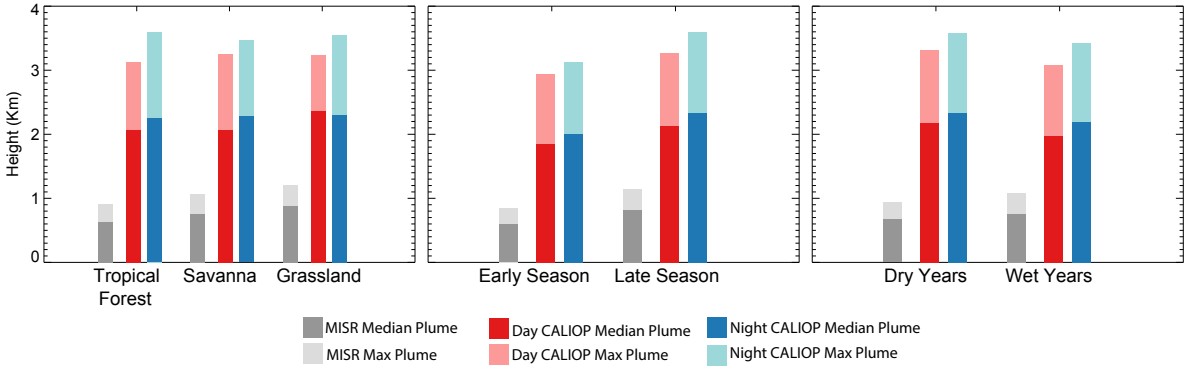

**Figure 10.** Average CALIOP and MISR plume heights per biome, moment of the season and dry/wet years. The burning season is divided into early (July–August–September) and later (October–November) periods, and dry years (2007, 2010) and wet years (2006, 2008, 2009, 2011). Bars represent MISR plume heights (grey), and daytime (red) and night-time (blue) CALIOP plume heights