# Peer review of "Biomass burning smoke heights over the Amazon observed from space"

_Atmospheric Chemistry and Physics, 2018_

## Short Comment (SC1) · 6 Nov 2018

Comment on "Biomass burning smoke heights over the Amazon observed from space" by Laura Gonzalez-Alonso et al.

Juliette Koppel

"Note to the editor and authors: As part of an introductory course to the Master programme Earth Environment at Wageningen University, students get the assignment to review a scientific paper. Since several years, students have been reviewing papers that are in open online discussion for Copernicus Journals, and the top students in the class have been asked to submit their reports to the discussion in order to help the review process. While these reports are written in the form of official (invited) re-

views, they were not requested for by the editor, and we leave it up to the editor and authors to use these reports to their advantage. We hope that these reports will positively contribute to the scientific discussion and to the quality of papers published. This report/review was supervised by Prof. Wouter Peters."

The goal of this research is to quantify the vertical distribution of fire smoke across the Amazon and to identify the key factors that control the plume height and rise. In order to achieve this goal, the smoke plume height and its variability will be characterized and the influences of different biome types, fire intensity, local atmospheric conditions and regional drought on smoke height will be studied. The climatology of 2005-2012 is limited for the burning seasons (July – November) and retrieved from space-borne observations from MISR and CALIOP. For all biomes there is a plume-height seasonal cycle and also for all biomes most smoke is located below 2 km. No clear relationship is found between drought conditions and fire radiative power. MISR and CALIOP show contradicting results regarding smoke plume heights and DSI, but CALIOP systematically detects higher smoke plumes than MISR. This work highlights the importance of biome type, fire properties and atmospheric conditions for plume dynamics, as well as the effect of drought conditions on smoke loading. The study demonstrates that combined observations of MISR and CALIOP allows for better constraints on the vertical distribution of smoke from biomass burning over the Amazon.

What is new in this paper is that there has not yet been any research on the vertical distribution of smoke plumes in the Amazon and also no research has yet been done on the key factors that influence the vertical distribution of fires. This research is of importance because of the great impact of Amazon fires on global biomass burning emissions. These emissions have a large influence on air quality, atmospheric composition, climate and ecosystem health. Therefore, it is necessary to gain a better insight in the vertical distribution of fires and the key factors influencing this process.

In my opinion, the paper is written very clear and has a good structure. The introduction is very strong, including societal significance, previous research, the reason

of the study area, the gap in research and good funnelling. In general, in the results/discussion section the results that are found are almost all compared with previous studies and explained well. The overall text is easy to read and written in a nice way so that the attention keeps to be drawn to reading the paper.

I think this paper fits well to the scope of the journal. The study is about smoke plumes present in the Earth's atmosphere and the underlying physical processes. One of the main research activities of the journal is Remote Sensing, which is in this paper is present in the method because of the use of MISR and CALIOP.

However, there are some sections in the paper that need to be revised in order to have this paper published. These adjustments are needed especially in regard to the methods of both MISR and CALIOP, the added value of using both MISR and CALIOP, the importance of land-management policies and some other minor aspects which I will elaborate on later in the review.

Major arguments

1) MISR and MODIS are both aboard on the NASA Terra satellite, which crosses the equator between 10:00 and 11:00 a.m. local time. This means that observations of smoke plumes will only be available for this time step every day. In this research also the smoke plume heights are related to boundary layer height and atmospheric stability. Specifically, this is done in the results/discussion section, page 9 line 25-34 and page 10 line 1-6.

In principle, stable boundary layer conditions occur when $\theta$(K)/Z(km) > 0 and unstable boundary layer conditions occur when $\theta$(K)/Z(km) < 0 (Vilà J., 2017-2018) . In the results and discussion section of this paper an atmospheric stability of <2 K/km is designated as weak and an atmospheric stability of > 4 K/km is designated as strong, see page 9 lines 33-34. But on what are these values based? All the MISR smoke plumes are categorized as having this weak or strong stability and results (further elaborated in the paragraph below) are based on this. The results can be doubted, since no ex-

planation is given for the criteria values of atmospheric stability and the values thus cannot be validated.

Figure 4 shows the vertical distribution of MISR plume height retrievals, classified under the weak and strong stability categories that are designated here. In lines 2-6, page 10 it is stated that "Our comparison supports previous observations that plumes under weak atmospheric conditions tend to inject smoke to higher altitudes than those experiencing strong stability, with average maximum plume heights of 1150 m and 654 m, respectively." It is also stated that same patterns are found for median and average plume heights. Another statement is that weak stability conditions are associated with deeper boundary layers than strong stability conditions, but it is also stated that this is not even shown. So, first of all, when the categories for weak and strong stability are not appropriately defined, this will cause non appropriate values for the percentage of plumes per category (presented on page 9 line 34 and in figure 4) and maximum, median and average plume heights per category as well (presented on page 10, lines 3-4. Second of all, since it is not even shown that deeper boundary layer heights are associated with weak stability conditions, this statement "Weaker atmospheric stability conditions are also associated with deeper PBLs (âĹij1500 m) than strong stability conditions (âĹij1200 m)." can't be made. On top of that, this very same statement is also a conclusion that is based on the weak/strong stability categories, so when these categories are not defined right, this statement might not even be true.

Furthermore, the MISR observations are only taken in the morning (10:00-11:00 local time) and thus all the conclusions regarding MISR observations that are made only gives us information for this time step. Since the boundary layer processes and height and atmospheric stability changes a lot during the day (Vilà J., 2017-2018), this time step might not be very representative. Information about the changing boundary layer processes during the day is missing in this paper, where I think it is necessary to include this specifically in the discussion section, page 9 lines 25-24. Also for the conclusions I think it should be stated clearly that this only accounts for the specific time step of

(10:00-11:00) and cannot be generalized for the day. In order to be able to test what the effect of changing atmospheric conditions during the day on plume height is, it is necessary to model (with for example model Daysmoke, Liu Y., et al 2010) the hourly PBL height and 6-hourly potential temperature profiles (obtained in this study) against the vertical distribution of smoke plumes.

2) In the paper it is stated at page 14, lines 5-7, that the initial objective of this research was to compare data from MISR with CALIOP. However, in the paper of Kahn et al., 2008 it is already stated that MISR and CALIOP observations are in fact complementary. Since this is known on beforehand and is mostly due to the properties of both instruments, I don't understand how the authors came to this initial objective. On top of that, in the abstract of the paper, page 1 lines 20-21, it is said that combined observations of MISR and CALIOP allows for better constraints on the vertical distribution of smoke from biomass burning over the Amazon. However, most conclusions in this research are based on the MISR data.

At page 7, lines 2-4, it is mentioned that for CALIOP, both day and night observations will be analysed, to allow a better comparison with the smoke plumes of MISR. But it is already known that comparison of observations of both instruments is not appropriate, and a cause of that is the difference in sampling time. This difference makes it even harder to compare data, because not the same smoke plumes are observed. This is also mentioned in the paper at page 14, lines 5-10. In the results section of the CALIOP smoke plume observations, it is found that the years with highest or lowest number of plumes are the same as observed by MISR and also the peak and biome type with highest biomass burning agree with MISR, page 13 lines 21-24. The only difference in smoke plume heights between MISR and CALIOP were that CALIOP observes smoke at systematically higher altitudes than MISR, stated at page 14 line 31, but this is also already found in previous studies. So for these results, CALIOP has no added value. Also, at line 25 page 14, it is stated that Huang et al., 2015 found the same smoke plume height values over the Amazon. Even though the method of Huang et al., 2015

is different, AOD is calculated for the whole Amazon area, while in this paper the AOD is calculated for individual plumes associated with active fires, no new information is found in this research. Maybe even Huang's results could have been used, because it could have been known that the individual plumes of CALIOP cannot be compared with MISR, so there is no added value in deriving them.

So it should be stated more clearly in the methods section of paper, why both instruments are being used in this research and in the results/discussion or conclusions section of the paper, what the additional value is of using both MISR and CALIOP instruments and not just MISR.

3) In the introduction at page 2 line 4, it is stated that land-management policies cause significant variability in (not mentioned clearly) the spatial variation of fires. After this, in the methods section at page 5 lines 15-16, it is also indicated that one of the years from the climatology (2006) is a year when land-management policies measured limited deforestation. Finally, in the conclusions section at page 16 lines 17-19, the paper states that strong land-management policies can become crucial for the Amazon in controlling fires with changing future climate conditions. Apparently, land-management policies are of importance regarding this research. However, even though one year of adjusted land-management policy is included in the climatology, nothing is mentioned about this in the results/discussion or in the conclusions section. This feels like a missed opportunity, because even though it is only one year in the climatology and maybe nothing significant is found, in the introduction, methods and at the end of the conclusion this research implies that land-management policies could influence biomass burning. Because of this I think this research should include some results or discussion points about this year in the research.

Minor arguments

Page 1, abstract/methods: It is nowhere explained why the dataset of MINX is 2005-2012 but the dataset of CALIOP is 2006-2012. CALIOP was launched in 2006, so

data of 2005 are impossible to obtain, but why does MINX also includes 2005 in the dataset? Please explain this in the methods.

Page 2, line 4: It is stated that significant variability exists. But it doesn't say between what aspects significant variability exists, so please indicate this more clearly in the text.

Page 3, lines 15-18: At the end of the introduction the objectives are mentioned. However what is missing here is the influence of land cover/biome type, because that is also studied in the paper. Please include this in the objectives.

Page 4, lines 13-14: The paper states that a user has to digitise the boundaries of the plume and indicate the direction of the smoke transport. How this should be done however, is not given in the paper. In order to be able to repeat the method I think it is necessary to indicate more clearly how the user should do this, or refer to a paper where this is done.

Page 5, lines 10-11: The best estimate maximum and median smoke plume heights are used, but it is not stated how these values are derived. In the paper of Martin M. V., et al 2010, the generation of these values is explained, but is it the same as for this study? And why are these two specific height definitions used and not the other ones that are given by MINX? Please explain this choice.

Page 5, lines 11-12: Smoke plumes are categorised with quality retrieval flags, but it is not explained how these categories are derived. The quality retrieval flags determine which plumes are taken into account for the climatology and which are not, so this could affect the total number of observations and it is important to have the right criteria for when a smoke plume should be qualified as good or bad. Thus it is important to be transparent about these quality retrieval flags, so please explain how these are derived.

Page 5, lines 23-25: In the paper it is said that the 60m difference in smoke plume heights between red and blue band retrievals can be neglected, because it is lower

than the MINX uncertainty of 250 m. However, when this difference is not negligible this might influence the results because not all observations are retrieved with red and blue band, some only with blue or red band. So also for this, it is important to explain clearly why this difference can be neglected and to add a reference for the MINX uncertainty.

Page 7, lines 14-15: Only the grid cells that contain at least two MODIS fire pixels are associated with active fires, at 80

Page 7, lines 22-24: To ensure there is no bias in the 0.5°x0.5° horizontal resolution, a 0.1°x0.1° horizontal resolution for 2017 is obtained and it is stated that there are no significant differences. But it is not stated clearly between what the differences are, please indicate this clearly.

Page 15/16, conclusion and summary: In my opinion there is not enough of a retrospect towards the reason of why this research has been of importance for the Amazon area. This is very well explained in the introduction and I think it would strengthen the conclusion section and the recommendation for further research, so please elaborate on this is the conclusion section.

Minor issues

Page 2, line 14: There seems to be a missing reference after the sentence: "The altitude...environmental impact", please include the source. Page 4, line 5: In this sentence there is referred to Kahn and Gaitley, 2015. However this reference is not given in the references section, please include this source. Page 4, line 24-33: This paragraph is about the limitation of the instruments and might be better for the discussion. Page 4, line 33: There seems to be a missing reference after the sentence: "In contrast...smoke layers", please include the source. Page 9, line 10: The word "of" is missing before the word "these". Page 13, line 4: The word "swallower" should be the word "shallower". Page 21, Table 2: Underneath the table there is some additional information where is referred to in the table with an "a" and a "b". However underneath the table there are two "a" and no "b", please change this. Page 23, Figure 2: The

time series that the figure is given for is not mentioned in the caption, please include this. Page 26, Figure 7: For MODIS FRP for the years 2007 and 2009 very high values are found, but nothing is said about this in the results. Also in this figure I don't really understand the necessary of putting the median value also in a number at the top of each boxplot, because it is already indicated inside the boxplot self. If there is no other reason behind putting this numbers here, then please remove them. Page 27, Figure 8: The symbols that are used for the years are hard to distinguish and difficult to interpret. Please use other symbols, or make them bigger, or find another way to indicate years.

References

Huang, J., Guo, J., Wang, F., Liu, Z., Jeong, M. J., Yu, H., Zhang, Z. (2015). CALIPSO inferred most probable heights of global dust and smoke layers. Journal of Geophysical Research: Atmospheres, 120(10), 5085-5100.

Kahn, R. A., Chen, Y., Nelson, D. L., Leung, F. Y., Li, Q., Diner, D. J., Logan, J. A. (2008). Wildfire smoke injection heights: Two perspectives from space. Geophysical Research Letters, 35(4).

Liu, Y., Achtemeier, G. L., Goodrick, S. L., Jackson, W. A. (2010). Important parameters for smoke plume rise simulation with Daysmoke. Atmospheric Pollution Research, 1(4), 250-259. Martin, M. V., Logan, J. A., Kahn, R. A., Leung, F. Y., Nelson, D. L., Diner, D. J. (2010). Smoke injection heights from fires in North America: analysis of 5 years of satellite observations. Atmospheric Chemistry Physics, 10(4).

Vilà, J. (2017-2018). Boundary Layer Processes: MAQ-32306 Lecture Notes. Wageningen University, Wageningen, Netherlands.

Please also note the supplement to this comment:
https://www.atmos-chem-phys-discuss.net/acp-2018-931/acp-2018-931-SC1-supplement.pdf

---

## Referee Comment (RC1) · Anonymous Referee #1 · 9 Nov 2018

General

The paper presents interesting but contradicting findings regarding Amazonian smoke layer heights retrieved from passive and active satellite remote sensing. Most parts of the paper are well written. However, some clarifications are needed.

As an example, we need precise wording throughout the article. We have 'smoke plume height'! What does that indicate: layer base, layer center, layer top? Only after checking the paper back and forth, it became clear to me what is meant. . . . . . For meteorologists, cloud height, for example, means cloud base height, in your case it probably means top height.

Regarding averaging. . .: Could be temporal and/or spatial (horizontal) averaging. . . so

be more specific, say clearly what you did! ...throughout the manuscript.

The conclusions must be improved! What can we do with these so different findings (active vs passive remote sensing).

Details:

P1 L7: you write ....1100m maximum plume height average... lowest plumes occur over tropical forest fires (800m). What do you mean here? What is the maximum plume height (is that related to layer top)? The lowest plumes occurred at 800m (intuitively that means layer base...) ...? Please improve this unprecise wording! ...throughout the entire Abstract! ...and the entire paper! And regarding averaging: you mean... spatial averaging, temporal averaging, or just avergaing of all cases?

P3, L8: There is this Baars et al. paper (JGR 2012), now mentioned in the introduction. This is the first systematic investigation of smoke layer geometrical and optical properties over an Amazonian site (a bit north of Manaus). You mention it, but you do not make any attempt to compare their results with yours. They measured smoke AODs with Raman lidar, they have measured lidar ratios, they have multiwavelength information for aerosol typing (fresh vs aged smoke etc), and layer base and top heights and depths for the fire season of 2008. But you use the much more uncertain CALIOP observations. In the case of CALIOP, the lidar ratio is more or less a look up table value, the CALIOP return signals are rather noisy, the CALIOP data analysis team even estimates the aerosol type from some kind of look up tables!

So my simple question is, why not using the Baars et al. (2012) results for comparison in addition?

By the way, this reviewer is not Dr. Baars, but an EARLINET Raman lidar specialist.

P4, L16, L18, L19, L21, ..........: Plume heights, yes I know, you mean plume top heights. Please, write that explicitly!

P5, L10: MINX computes several plume heights... you mean....top heights....

P5, L11: We use maximum and median smoke plume heights... Top heights? Median heights... regarding.... the entire season of a year, for the entire region you cover with observations??? Just all plumes, you collected???

P6, section 2.6: You concentrate on the comparison with CALIOP observations! How is the comparison of CALIOP with the Baars 2012 results for the fire season of 2008 regarding layer base and top heights, aerosol typing, lidar ratios?

P7: again precise wording is necessary...

P8, results and discussion sections 3: I would like to see a 1:1 case study, with a CALIOP fire smoke profile with indicated base height, center height, and top height, and then what you got from your MISR retrieval ... as layer top height (even if the measurments are done at very different times of the day and PBL evolution...). This would provide better grounds to discuss the huge discrepancies between passive and active remote sensing products regarding smoke layer tops.

P14, L10: 'complementary' What is complementary when the CALIOP and MISR products are so much different?

P14, L30: Nice to have all these references from very different regions. But the main question remains: What did Baars et al. (2012) report for the Amazonian forest in the Manaus area? And how does that fit into the picture seen by MISR and CALIOP?

P15, P16: At the end what is now the conclusion, having these huge discrepancies between spaceborne lidar and passive remote sensing lidar in mind?

I am lost after the discussion, and even after reading the conclusions. How to proceed with this? How can modellers make use of such contradicting MISR/CALIOP results?

Figure 1: Yes, I am a lidar scientist, but nevertheless, I had trouble to understand the text in the figure captions: smoke plume median heights? What does that mean here? There are then two color scales, what belongs to what? Yes at the end, I got it after minutes of 'research'. Colored circles for different aerosol types: green for dust, up to

12 km, really? Any idea about the dust source? Next: Dashed black line represents the averaged extinction profile (??) What did you average, and why is that a function of height? So, smoke is indicated by pink dots! Fine! But there seem to be a lot of clean/continental air parcles on 25 Sep, scattered all over the insert display, even at dust level heights of 10-12 km? Confusing! ... but understandable. The aerosol typing is based on questionable CALIOP look up tables!

Figure 9: What did Baars et al. (2012) observe in 2008?

Figure 10 shows the final result!

... and my personal spontaneous conclusion and main question after reading the entire manuscript is: Having these huge differences in the findings in mind, what is then complementary (after analysing CALIOP and MISR smoke observations)? How should modellers (most are not experts of passive and active remote sensing) use the 'combined' information? Can we, e.g., quantify ... from the combined observations... how much smoke AOD is in the layer below the MISR-derived top height, what is the residual AOD for the layer between MISR and CALIOP-derived top heights?

Please, explain that in the conclusion section what is now the concrete result of this work. How can we use these data sets...? What is the true information content. Many readers will not be familiar with passive or active remote sensing, but are interested in Amazonian fire smoke and the horizontal and vertical distribution, and potential consequences for long range transport and deposition.... Please help them to understand the findings.

I like the results! Many authors would hesitate to show us the 'real world' of observations, retrievals, and apparently contraditing products. I think it will not be so much work to revise the mansucript a bit to meet (some of) my points.

---

## Referee Comment (RC2) · Anonymous Referee #2 · 22 Nov 2018

The study by Gonzalez-Alonso et al. explores the injection height of biomass burning emissions across the Amazon during the dry season and produces a climatology of smoke plumes heights derived satellite observations. Overall the paper is well written and concepts are described clearly (although some sections in the methods may benefit from a summary figure or table – see comments below). Figures are really nicely displayed and, mostly, very easy to understand. The topic of the study is well within the scope of ACP and I can see that this dataset will be very useful, particularly to modellers simulating processes & impacts of biomass burning in the Amazon region. I recommend publication once the comments below have been addressed.

General

1. The methodology explanation is very thorough and well written. However, the meth-

ods section is very long and sometimes a bit hard to follow, particularly for readers that are unfamiliar with these satellite products. I suggest adding a table or two (or extending Table 1) summarising the main datasets and tools used including information on the satellite products and the version used, resolution, overpass time etc. The methods section would also benefit from a schematic diagram perhaps just of the MINX software, to make the analysis process clearer.

2. In the paper, the authors make a good attempt to compare parts of the methods and results to previous studies. However, these sections are buried in the text. I wonder if it would make the manuscript clearer if you had a separate section where you compare the methods and results to previous studies? You could add a table including previous studies on plume heights in the Amazon either using similar or different methods (e.g. Baars et al. 2012; Marenco et al., 2016), summarising/comparing the findings of these studies and yours.

3. I agree with Referee #1, the results from CALIOP and MISR are so different that I believe the reader will be left feeling a bit unsure of what information to take from the paper (particularly atmospheric chemistry/aerosol modellers who may not be familiar with the details of these satellite products/tools). Can you make some recommendations in the conclusions?

4. I strongly suggest comparing your results to what is currently used/assumed in atmospheric aerosol/chemistry models for fire emissions injection heights. Do your results contradict or confirm what is currently used?

5. Following on from the previous general comments, for modellers it would be extremely useful to have idea of whole vertical distribution of the plumes rather than just median/max plume height. For example, the average percentage of the plume in specified altitude bands. Could this information be estimated from the CALIOP data? Or perhaps this would be unreliable given the large difference between CALIOP and MISR results?

Specific

1. Abstract, L2: Specify the dry season months.

2. Abstract, L4-6: Sentence not written very clearly "About 60% of smoke plumes are observed during drought years, at the peak month of the burning season (September; 40–50%) and over tropical forest and savanna regions (94%)." Does this mean: 60% of smoke plumes were observed in drought years (relative to non-drought years); 40-50% observed in the peak month of burning season (relative to the other months); and 94% observed over tropical and savannah regions (with the remainder over grassland)?

3. Introduction: Order citations correctly (by year).

4. Introduction, P2, L12-14: Can you include any refences for why altitude to which smoke is injected is critical? Perhaps give examples of modelling studies where this has been tested e.g. some of the SAMBBA modelling papers, or observational studies.

5. Introduction, P3, L1-3: The Kolusu et al. (2015) and Thornhill et al. (2017) papers are modelling studies not observational (correct this sentence).

6. Introduction, P3, L9-10: "…no analyses yet that seek to quantify the vertical distribution of smoke from fires across the Amazon…" Suggest changing to: "…quantify the long-term average vertical distribution".

7. Sect. 2.1: How are the MISR and MINX vertical resolutions accounted for? Apologies if this explained later in the manuscript.

8. Sect. 2.6: Why was 0.5x0.5 degrees resolution chosen for CALIOP?

9. Sect. 2.6 (P7, L26-35 – P8, L1-5): Nice explanation of the other CALIOP products that have been used by previous studies. This may be helped by a table summarising: the studies (including yours), different products used, region studied etc. Also, is it possible to briefly say what the implications are for using these different products and explain why the specific product and plume height definition were chosen for your study

over the others?

10. This sentence from Sect. 3.1: "The majority of the plumes in this record are digitised with blue band retrievals (Table 1)...", seems to contradict the following two from Sect. 2.2 (or at least have confused me): "This screening leaves a total of 5393 plumes, about 56% of the original database, with 77% and 23% plumes digitised in the red and blue bands, respectively.", and "In our dataset overall, most of plumes are digitised with red band images, as it was the default option for MINX v2–3."

11. Be consistent with use of "PBL"/ "BL".

12. P12, L7 & L17: What are the p-values and at what confidence level is the relationship statistically significant?

13. P13, L15-16: "Our results suggest that fires during drought periods might significantly degrade regional air quality, as they are associated with low smoke altitude and large aerosol loading.". The finding that drought periods are associated with large aerosol loadings, which substantially degrade regional air quality is consistent with Reddington et al. (2015) (and other studies?). The higher aerosol loadings are likely due to the increases in the number/size of fires (e.g. Aragão et al., 2007; 2014) and subsequent increases in aerosol emissions. However, the potential for lower smoke altitudes in drought years, I'm assuming, is a new finding and should be highlighted/made clearer.

14. P14, L18-19: Would it not be possible to check the PBL height around CALIOP overpass time with MERRA2 data?

15. I'm not sure I understand how Figure 8 demonstrates the following statements: - P12, L7-9: "We find a significant positive relationship between MISR maximum plume heights and MODIS DSI (r=0.7) in tropical forest and savanna fires, with higher maximum plume heights in wet (1000–1100 m) than severe drought conditions (800–900 m) (Figure 8)." - P13, L3-4: "...tropical forest fires inject a larger percentage of smoke

plumes into the FT in wet than extreme-dry conditions (12 versus 20%, Figure 8)". Since I can only see data points for tropical forest in "Extreme- Severe" and "Mild-Moderate" conditions, with one point in "Normal" and none in "Wetter than Normal".

Tables & Figures

1. Table 1: suggest adding a "total" row in the table.

2. I suggest adding a table to summarise smoke plume heights for the main biomes (could also add drought/non-drought year averages) to compliment Figure 10. So that readers can get this info quicker than reading it off a figure.

3. Figure 2: I suggest adding one or two figures (perhaps to supplementary) to show the month and/or year the plume occurred e.g. with different colours.

4. Figure 10: If the differences between Day and Night CALIOP Median Plumes are not significant then is it worth just combining these here (keeping the separation in the previous figure)? It is really the difference between the CALIOP and MISR estimations of plume heights that is the significant result.

References (not already included in the paper)

Aragão, L. E. O. C. et al. Environmental change and the carbon balance of Amazonian forests. Biol. Rev. 89, 913–931 (2014).

Aragão, L. E. O. C. et al. Spatial patterns and fire response of recent Amazonian droughts. Geophys. Res. Lett. 34, L07701 (2007).
* * *

---

## Referee Comment (RC3) · Anonymous Referee #3 · 3 Dec 2018

General Comment:

In this manuscript, the authors characterize burning biomass plume height over the Amazon using MISR and CALIOP observations. They investigate the effect of FRP, atmospheric stability and drought while considering seasonal and interannual variabilities. This is the first time that such work was performed over the Amazon. The manuscript is well structured and well written. The discussion on the drought is particularly interesting. I would recommend this manuscript for publication in ACP after considering the comments listed below. The important point that need to be addressed in the correction is the definition and the use of FRP that cannot be directly linked to fire intensity (see third comment below).

Specific comments:

[Figure]

P2 L3: consider including in the text some geographical location of where we should materialize this arc of deforestation.

P2 L11-14: consider referencing the review on plume injection height from Paugam et al. 2016 when discussing the effect of plume injection height.

P2 L21: fire intensity is a specific metrics in fire science expressed in [W/m] and is not the same as FRP[W] or FRP density [W/m2]. However, when dealing with satellite observation, FRP density is usually related to fire intensity. Your definition of fire intensity should be discussed at this point in the introduction. You use in the remaining of the manuscript FRP as a metric for fire intensity. FRP is an estimate of the total radiant energy emitted by the active surface area of the fire, flaming and smoldering area all included. FRP is probably better defined as a measure of fire activity including size and radiant heat flux (FRP density).

P3 L11: capital letter for Fire Radiative Power (FRP). You could add the MODIS collection version here.

P3 Section2.1: Consider grouping the paragraph on MISR and MODIS, ie l21 to 28 could be moved to the start of page 4.

P3 L32: see comment above on FRP and fire intensity.

P5 L11: replace "),in" to "), in"

P5 L20-23: consider moving the discussion on the red and blue band in the Supplementary Information. As you showed the added error is negligible compared to the MINX uncertainty. You could just mention it once and refer to the SI for more details.

P6 L2: consider also mentioning here when you consider an atmosphere stable or not.

P7 L2: "wide range of condition as in MISR". I am not sure I understand why your methodology is ensuring a wide range of condition.

P5 paragraph 3: I would move this section after you mention the choice of your horizontal resolution (line 5).

P5 paragraphs order: Consider rearranging the paragraph order in this page to make it easier for the readers. For example, you mention twice how you define CALIOP plume height. This is only a suggestion: the last paragraph on the definition of CALIOP plume height should come after you first mention how you define plume height (line 8). Then would come the discussion on how you link the plume to fire activity.

P5 L18: why do you expect a bias?

P5 L29: Most Probable Height.

P5 L10: consider mentioning that "those grid cells" are the grid cells of your gridded CALIOP injection height product.

P5 L10: How do you cluster MODIS Fire pixels? Are you taking the larger cluster or do you sum all fire pixel in the grid?

P5 L12: Are you using the same elevation model than in MISR?

P8 L15-16: as mention above, I think this is not brining any added value to the discussion here. Move the discussion on MINX band retrieval in the SI.

P9 L5-6: Could you mention how does this stability metric relate to the definition of the stability flag (stable/unstable) defined in Val Martin et al 2010? I might miss the point, but why do you define a new metric as you seem to only define atmosphere as weakly and strongly stratified as in Val Martin et al 2010.

P9 L8: "a summary of these"

P9 L14: your value of FRP contrast with the FRP density values reported by Freitas et al 2007 for the same biodomes. Grassland is reported to have an FRP density (3.3 kW/m2) an order of magnitude lower than tropical forest (30-80 kW/m2).

P9 L21: "obscuring the fire emitted 4-micron radiance [. . .] as well as low radiant emissivity". Consider reformulating this sentence. Why the flame emissivity should alter the FRP retrieval in tropical forest? The FRP formulation relies on the gray body assumption. Flaming combustion (because of soot presence in the flame) is more prone to violate the gray body assumption than smoldering. In case of smoldering fire, vegetation absorption is more likely to alter FRP estimate.

P9 L25: you could mention that some simulation studies also work on the impact of atmospheric stability and that this is still an open problem in plume rise parameterization. The plume rise model proposed in Paugam et al 2015 (based on the original work of Freitas et al 2007) was shown to be sensitive to atmospheric stability unlike others existing parameterizations. However, this work was refused for publication in ACP, and despite this publication refusal, results of the same model implemented in GFAS were published in ACP in Remy et al 2017.

P9 L27: consider reformulating: "and weaker atmospheric stability conditions when low altitudes plumes then to be trapped with the boundary layer".

P9 L31: as mentioned above, why not using the same flag as in Val Martin et al 2010 to define the state of the atmosphere.

P9 L32: Figure 4

P10 L12-13: I am not sure this sentence brings much to the discussion. Consider removing it.

P10 L20-21: combustion efficiency is probably more related to FRP density than FRP. Active fire area is important in your discussion here and should be mentioned.

P10 last paragraph: AOD corelate also to the FRP time integration (= Fire Radiative Energy, FRE), see Pereira et al 2009.

P11 L17: why smaller fire in size require less conservative definition in FT injection?

P11 L24-27: I found the discussion slightly confusing. Does the height the PBL relates

to the strength of the stable layer located just above? I might be wrong this is just a thought. In the presence of a deep PBL, there might have quite a lot of water vapor that could be used by the convective plume to get stronger, get across the stable layer and reach the FT.

P12 L1: "as discussed above". Mention the section.

P12 L2-3: "Note that DSI is higher in wetter years." Is this not just the definition?

P12 L5: than in severe drought condition.

P12 L23-30: I found the discussion difficult to read. If I well understood your point is that drought effects are correlated with the biodome of their geographical location. Drought between 2005 and 2007 move from one biodome to another. Could you just discuss FRP and injection height changes for the two biodomes between the two years? Why are you using in this discussion the repartition of all observed plume per biodome (Fig S1)?

P21 L32: what are the mechanisms that make a PBL deeper in dry year?

P13 L13: However, you mentioned that grassland fire might reach higher injection height in dry condition?

P14 L17: According to what your argumentation in section 3.4, regardless of PBL, tropical forest fires plumes are lower in dryer condition. So your point here only applies to grassland fire?

P15 L9: "more opportunity to mix upward". MISR data shows generally a peak injection height near the fire where the convective plume is active (with potentially pyroconvection taking place) and then a downdraft caused by the aerosol loading and the atmospheric stratification. A later updraft is possible on longer time scale for older plume through solar radiation heating (De Laat 2012). I think that the main processes responsible of the differences between plume smoke observed by MISR and CALIOP are changes of atmospheric stability and fire activity which can make the updraft core

of the plume stronger, making aerosol spreading at higher altitude. Aerosol that were emitted earlier in the day would not have time to reach higher altitude just by solar heating.

P26 L9-14: as already mentioned, the discussion on fire intensity would be better related to FRP density rather than FRP.

References:

Pereira, G., Freitas, S. R., Moraes, E. C., Ferreira, N. J., Shimabukuro, Y. E., Rao, V. B., & Longo, K. M. (2009). Estimating trace gas and aerosol emissions over South America: Relationship between fire radiative energy released and aerosol optical depth observations. Atmospheric Environment, 43(40), 6388–6397. https://doi.org/10.1016/J.ATMOSENV.2009.09.013

de Laat, A. T. J., Stein Zweers, D. C., Boers, R., & Tuinder, O. N. E. (2012). A solar escalator: Observational evidence of the self-lifting of smoke and aerosols by absorption of solar radiation in the February 2009 Australian Black Saturday plume. Journal of Geophysical Research: Atmospheres, 117(D4), n/a-n/a. https://doi.org/10.1029/2011JD017016

Paugam, R., Wooster, M., Freitas, S., & Val Martin, M. (2016). A review of approaches to estimate wildfire plume injection height within large-scale atmospheric chemical transport models. Atmospheric Chemistry and Physics, 16(2), 907–925. https://doi.org/10.5194/acp-16-907-2016

Paugam, R., Wooster, M., Atherton, J., Freitas, S. R., Schultz, M. G., & Kaiser, J. W. (2015). Development and optimization of a wildfire plume rise model based on remote sensing data inputs – Part 2. Atmospheric Chemistry and Physics Discussions, 15(6), 9815–9895. https://doi.org/10.5194/acpd-15-9815-2015

Freitas, S. R., Longo, K. M., Chatfield, R., Latham, D., Silva Dias, M. A. F., Andreae, M. O., . . . Carvalho, J. A. (2007). Including the sub-grid scale plume rise of vegetation fires

in low resolution atmospheric transport models. Atmospheric Chemistry and Physics, 7(13), 3385–3398. https://doi.org/10.5194/acp-7-3385-2007

Rémy, S., Veira, A., Paugam, R., Sofiev, M., Kaiser, J. W., Marenco, F., Burton, S. P., Benedetti, A., Engelen, R. J., Ferrare, R., and Hair, J. W.: Two global data sets of daily fire emission injection heights since 2003, Atmos. Chem. Phys., 17, 2921-2942, https://doi.org/10.5194/acp-17-2921-2017, 2017.

---

## Author Comment (AC1) · 8 Jan 2019

**Response to Reviewer #1**

We thank the anonymous reviewer for her/his thorough evaluation and constructive recommendations for improving this manuscript. Her/his comments (in italics) and our responses are listed below.

*General comments*

*The paper presents interesting but contradicting findings regarding Amazonian smoke layer heights retrieved from passive and active satellite remote sensing. Most parts of the paper are well written. However, some clarifications are needed. As an example, we need precise wording throughout the article.*

*We have 'smoke plume height'! What does that indicate: layer base, layer center, layer top? Only after checking the paper back and forth, it became clear to me what is meant...... For meteorologists, cloud height, for example, means cloud base height, in your case it probably means top height.*

For MISR, we report the elevation above the geoid of the level of maximum spatial contrast in the multi-angle imagery. This is generally near the plume top, but it actually provides a distribution of heights in most cases, because aerosol plumes are rarely uniform. The centroid of this distribution is typically somewhere within the plume (e.g., Fig. 2 in *Flower and Kahn*, J. Volcanology, 2017). On the other hand, CALIOP tends to report higher plume-height when very thin aerosol, to which the lidar is more sensitive, resides above the main plume deck. We have reworded the manuscript to clarify the definition of smoke plume height. We added this information in Sections 2.1 and 2.2.

Page 4 lines 24-27

> [..] MINX plume heights are reported above the geoid, which correspond to the level of maximum spatial contrast in the multi-angle imagery, typically near the plume top, but actually offering a distribution of heights in most cases, because aerosol plumes are rarely uniform (*Flower and Kahn*, 2017). Additionally, MINX provides local terrain height from a digital elevation map (DEM) product.

Page 5 lines 17-19

> MINX computes several plume heights that describe the altitude that smoke reaches in the atmosphere. In this work, we use the best estimate maximum and median smoke plume heights, which represent the distribution of stereo heights, obtained at the level of maximum spatial contrast over the plume area [Nelson et al 2013].

*Regarding averaging...: Could be temporal and/or spatial (horizontal) averaging... so be more specific, say clearly what you did! ...throughout the manuscript.*

We edited the manuscript to make clearer what our averages refer to.

*The conclusions must be improved! What can we do with these so different findings (active vs passive remote sensing).*

The multi-angle and lidar techniques are sensitive to different aspects of plume height, and are essentially complementary (e.g., Kahn et al., 2008). As also suggested by reviewer 2, we have added more information in the Conclusions to make our findings clearer.

*Details:*

*P1 L7: you write ....1100m maximum plume height average... lowest plumes occur over tropical forest fires (800m). What do you mean here? What is the maximum plume height (is that related to layer top)? The lowest plumes occurred at 800m (intuitively that means layer base...) ...? Please improve this unprecise wording! ...throughout the entire Abstract! ...and the entire paper! And regarding averaging: you mean...spatial averaging, temporal averaging, or just avergaing of all cases?*

As discussed above we have reworded the manuscript to make clearer what smoke plume heights mean and what our averages refer to.

*P3, L8: There is this Baars et al. paper (JGR 2012), now mentioned in the introduction. This is the first systematic investigation of smoke layer geometrical and optical properties over an Amazonian site (a bit north of Manaus). You mention it, but you do not make any attempt to compare their results with yours. They measured smoke AODs with Raman lidar, they have measured lidar ratios, they have multiwavelength information for aerosol typing (fresh vs aged smoke etc), and layer base and top heights and depths for the fire season of 2008. But you use the much more uncertain CALIOP observations. In the case of CALIOP, the lidar ratio is more or less a look up table value, the CALIOP return signals are rather noisy, the CALIOP data analysis team even estimates the aerosol type from some kind of look up tables!*

*So my simple question is, why not using the Baars et al. (2012) results for comparison in addition?*

*By the way, this reviewer is not Dr. Baars, but an EARLINET Raman lidar specialist.*

We thank the reviewer for pointing to Baars et al., (2012), which had initially been overlooked in our first version of the manuscript. We carefully thought about the reviewer's suggestion about comparing our results with Baars et al (2012). Our manuscript presents a comprehensive climatology (2007-2012) of smoke plume heights retrieved from CALIOP over the entire Amazon domain, whereas Baars et al, (2012) cover year 2008 and at one specific point location (2.5ºS, 60ºW). On the other hand, and as the reviewer mentions, Baars et al (2012) presents a more detail analysis of smoke layer geometrical and optical properties. We feel that mentioning results from Baars et al (2012) with respect to smoke plume height and aerosol loading, which we do, is appropriated for the scope of our paper.

*P4, L16, L18, L19, L21, .. ... ....: Plume heights, yes I know, you mean plume top heights. Please, write that explicitly!*

We mean as plume height the level of maximum spatial contrast in the multi-angle views, not the mean plume top, as explained above. We clarified this point in the manuscript as discussed above.

*P5, L10: MINX computes several plume heights... you mean....top heights.*

Clarified as discussed above.

*P5, L11: We use maximum and median smoke plume heights... Top heights? Median heights... regarding.... the entire season of a year, for the entire region you cover with observations??? Just all plumes, you collected???*

These statistics represent the distribution of stereo heights, obtained at the level of maximum spatial contrast over the plume area, and stratified by season, year, etc., as appropriate. We have clarified it as discussed above.

*P6, section 2.6: You concentrate on the comparison with CALIOP observations! How is the comparison of CALIOP with the Baars 2012 results for the fire season of 2008 regarding layer base and top heights, aerosol typing, lidar ratios?*

Discussed above.

*P7: again precise wording is necessary...*

Reworded as discussed above.

*P8, results and discussion sections 3: I would like to see a 1:1 case study, with a CALIOP fire smoke profile with indicated base height, center height, and top height, and then what you got from your MISR retrieval ... as layer top height (even if the measurements are done at very different times of the day and PBL evolution...). This would provide better grounds to discuss the huge discrepancies between passive and active remote sensing products regarding smoke layer tops.*

The differences between CALIOP and MISR can be large in some case, but they are not huge, and they are consistent with the differences in overpass time and sensitivities of each measurement to actual aerosol plumes. These two sensors complement each other as explained above. We have made this point clearer within the manuscript. For example, we moved and expanded the discussion of CALIOP/MISR differences to Section 2.6 (Methodology), so the readers can learn about the differences and complementarities of these two satellite products before facing the results.

Page 8 Lines 16-23

> Our initial objective was to compare the CALIOP with the MISR plumes to assess the diurnal smoke evolution on a plume basis, as CALIOP has a later sampling time than MISR over the Amazon (13:00–15:00 LT versus 10:00–12:00 LT). However, despite our effort to develop a comprehensive CALIOP climatology none of the CALIOP plumes coincide with the MISR plumes. As previous studies discuss (e.g., Kahn et al., 2008; Tosca et al., 2011), CALIOP and MISR, in addition to having different sampling times, also have different swath widths (380 km versus 70 m). These differences make it difficult to observe the same fire on the same day, but they make CALIOP and MISR observations complementary: MISR provides late-morning near-source constrains on aerosol plume vertical distribution, whereas CALIOP in general offers more regional constrains, later in the day (Kahn et al., 2008). Some differences between the products are thus expected.

Reviewer suggests to show a comparison MISR-CALIOP on a plume basis. That was our initial intent but, given the differences in swath widths and temporal coverage, that is not possible. We have moved this discussion to section 2.6 (page 8, lines 16-23) as mentioned above. In addition, our Figure 1 provides the CALIOP fire smoke profile that reviewer would like to see. We have modified the caption to make Figure 1 clearer as discussed below.

*P14, L10: 'complementary' What is complementary when the CALIOP and MISR products are so much different?*

MISR provides extensive near-source mapping, whereas CALIOP provides downwind sampling. This is the subject of Kahn et al., (2008). We have added a discussion on the manuscript to clarify this point as mentioned above.

*P14, L30: Nice to have all these references from very different regions. But the main question remains: What did Baars et al. (2012) report for the Amazonian forest in the Manaus area? And how does that fit into the picture seen by MISR and CALIOP?*

We have modified the discussion to put results from Baars et al (2012) into context.

Page 15 Lines 3-8

> Smoke plume height values over the Amazon similar to ours were reported in other studies for CALIOP (Huang et al. 2015) and surface-based lidar measurements (Baars et al 2012). Using the CALIOP vertical feature mask and AOD profiles, Huang et al. (2015) reported an average for the most probable smoke height of 1.6–2.5 km for September fires. Their definition is comparable to our CALIOP median plume height, which produced a value of 2.3±0.7 km for the September months. Over Manaus in 2008, Baars et al., (2012) reported biomass burning layers at 3-5 km elevation, with most of the smoke trapped below 2 km.

Page 15 Lines 10-13

> In our study, CALIOP observes smoke at systematically higher altitudes than MISR, with median plume heights up to 1.4 km higher (2.2 km for the maximum plume heights). However, CALIOP still shows that the majority of the smoke is located at altitudes below 2.5 km above ground, consistent with previous observations from lidar measurements (Baars et al., 2012).

*P15, P16: At the end what is now the conclusion, having these huge discrepancies between spaceborne lidar and passive remote sensing lidar in mind? I am lost after the discussion, and even after reading the conclusions. How to proceed with this? How can modellers make use of such contradicting MISR/CALIOP results?*

We disagree with the reviewer. We do not find "huge discrepancies" between CALIOP and MISR. Differences in sampling, and in what each technique is actually sensitive to, explain the differences. Such differences are not discrepancies. We have reworded the manuscript, including Conclusions, to make the MISR-CALIOP comparison clearer, as mentioned above.

*Figure 1: Yes, I am a lidar scientist, but nevertheless, I had trouble to understand the text in the figure captions: smoke plume median heights? What does that mean here? There are then two color scales, what belongs to what? Yes at the end, I got it after minutes of 'research'. Colored circles for different aerosol types: green for dust, up to 12 km, really? Any idea bout the dust source? Next: Dashed black line represents the averaged extinction profile (??) What did you average, and why is that a function of height? So, smoke is indicated by pink dots! Fine! But there seem to be a lot of clean/continental air particles on 25 Sep, scattered all over the insert display, even at dust level heights of 10-12 km? Confusing! ... but understandable. The aerosol typing is based on questionable CALIOP look up tables!*

We reworded the caption to make the figure easier to interpret.

Page 25, Figure 1

Example of the approach followed for the CALIOP smoke plume characterisation. The map shows estimated smoke plume median heights (gridded at 0.5x0.5 horizontal resolution) for September 25th, 2010 at 06:25 UTC. MODIS active fire pixels associated with the CALIOP smoke plumes are represented with open circles. The insert displays the vertical distribution of aerosol extinction for a specific smoke plume in the map, with extinction values coloured by classified aerosol types. Dashed black line represents the averaged extinction profile for the aerosols classified as smoke (pink dots). In this profile, the CALIOP smoke plume has a median height of 1.7 km (green color in the smoke plume median height scale) and a maximum height of 4.5 km above the terrain

The dust at 12 km is most likely transported from North Africa. There is a vast literature about Saharan dust transport to the Amazon, e.g., Yu et al., (2015), Ben Ami et al., (2010).

*Figure 9: What did Baars et al. (2012) observe in 2008?*
Added a discussion on results from Baars et al (2012) as mentioned above.

*Figure 10 shows the final result!... and my personal spontaneous conclusion and main question after reading the entire manuscript is: Having these huge differences in the findings in mind, what is then complementary (after analysing CALIOP and MISR smoke observations)? How should modellers (most are not experts of passive and active remote sensing) use the 'combined' information? Can we, e.g., quantify ... from the combined observations... how much smoke AOD is in the layer below the MISR-derived top height, what is the residual AOD for the layer between MISR and CALIOP-derived top heights?*

*Please, explain that in the conclusion section what is now the concrete result of this work. How can we use these data sets...? What is the true information content. Many readers will not be familiar with passive or active remote sensing, but are interested in Amazonian fire smoke and the horizontal and vertical distribution, and potential consequences for long range transport and deposition.... Please help them to understand the findings.*

*I like the results! Many authors would hesitate to show us the 'real world' of observations, retrievals, and apparently contradicting products. I think it will not be so much work to revise the mansucript a bit to meet (some of) my points.*

We think that these results have been shown throughout the manuscript. We have emphasized the key points as described above. Although both the lidar and multi-angle imagery measure some aspect of aerosol plume elevation, they do not measure the same thing. We have clarified this in the manuscript as discussed in detail above. The height differences shown in Figure 10 are not that large, given the differences in the sensitivities and sampling of these techniques (1 vs. 2 or 3 km). Most of the plumes are likely within the PBL. We have reworded the manuscript, including the Conclusions, to make clearer our MISR-CALIOP results. In addition, as indicated by reviewer 2, we included an analysis of the PBL heights at the time of the CALIOP overpass, which help explain some of the MISR/CALIOP differences.

---

## Author Comment (AC2) · 8 Jan 2019

**Response to Reviewer #2**

We thank the anonymous reviewer for her/his thorough evaluation and constructive recommendations for improving this manuscript. Her/his comments (in italics) and our responses are listed below.

*The study by Gonzalez-Alonso et al. explores the injection height of biomass burning emissions across the Amazon during the dry season and produces a climatology of smoke plumes heights derived satellite observations. Overall the paper is well written and concepts are described clearly (although some sections in the methods may benefit from a summary figure or table – see comments below). Figures are really nicely displayed and, mostly, very easy to understand. The topic of the study is well within the scope of ACP and I can see that this dataset will be very useful, particularly to modellers simulating processes & impacts of biomass burning in the Amazon region. I recommend publication once the comments below have been addressed.*

We thank the reviewer for these kind words.

*General Comments*

*1. The methodology explanation is very thorough and well written. However, the methods section is very long and sometimes a bit hard to follow, particularly for readers that are unfamiliar with these satellite products. I suggest adding a table or two (or extending Table 1) summarising the main datasets and tools used including information on the satellite products and the version used, resolution, overpass time etc. The methods section would also benefit from a schematic diagram perhaps just of the MINX software, to make the analysis process clearer.*

As suggested by the reviewer, we have added a table summarizing the products and instruments used in our study. To avoid making the paper longer, we included this table in Supplementary Materials (Table S1) and added a reference in the text.

Page 3 lines 25-26

> "We provide below a summary of main datasets and tools used in the analysis and compile their main features in Table S1 (Supplementary Materials)."

We thank the suggestion about the MINX diagram, but we feel that there is already a significant amount of information in the literature about MISR smoke plume heights, e.g., Val Martin et al. (2010, 2018), Tosca et al. (2011), Mims et al. (2010), and the MINX software (Nelson et al., 2013), which we cite extensively throughout our manuscript, in particular Nelson et al. (2013). We prefer to refer the readers to these previous works.

*2. In the paper, the authors make a good attempt to compare parts of the methods and results to previous studies. However, these sections are buried in the text. I wonder if it would make the manuscript clearer if you had a separate section where you compare the methods and results to previous studies? You could add a table including previous studies on plume heights in the Amazon either using similar or different methods (e.g. Baars et al., 2012; Marenco et al., 2016), summarising/comparing the findings of these studies and yours.*

We thank the reviewer for this suggestion. We do compare different methods in the text and figures, as appropriate; however, we think that there are not that many studies on smoke plume heights over the Amazon to justify a separate section.

*3. I agree with Referee #1, the results from CALIOP and MISR are so different that I believe the reader will be left feeling a bit unsure of what information to take from the paper (particularly atmospheric chemistry/aerosol modellers who may not be familiar with the details of these satellite products/tools). Can you make some recommendations in the conclusions?*

We have reworded the manuscript to make clearer how results from MISR and CALIOP are complementary and, given differences in the sensitivity and sampling of these techniques, not so different.

In addition, as suggested by the reviewer later in Point 14, we extracted the PBL heights at the location and time of the CALIOP smoke plumes. As briefly mentioned in our manuscript, PBL is expected to grow further later in the day at the time of the CALIOP observation. We find that PBL at the time of the CALIOP daytime overpass (2-3 pm LT) is about 1.4 km deeper than at the MISR overpass time (10-11 am LT). A deeper PBL contributes to the difference observed between MISR and CALIOP smoke plumes. Fires can also become more energetic as the day wears on, increasing plume buoyancy and smoke injection height. This is an important point that we did not emphasize in our first version and we do it now. We made the following changes:

Page 14 line 33 - page 15, line 2

> […] In contrast, CALIOP observes smoke at higher altitudes during dry (2.2 and 3.4 km) than wet years (2.0 and 3.2 km). As discussed above, for the time and location of the MISR observations, a deeper PBL is observed in dry compared to wet years. Likewise, PBL heights at the CALIOP smoke plumes are 2.4 and 2.6 km in wet and dry years, respectively, and thus a deeper PBL during drought conditions explain the higher altitudes observed by CALIOP under drier conditions.

Page 15 lines 10-16

> In our study, CALIOP observes smoke at systematically higher altitudes than MISR, with median plume heights up to 1.4~km higher (2.2 km for the maximum plume heights). However, CALIOP still shows that the majority of the smoke is located at altitudes below 2.5 km above ground, consistent with previous observations from lidar measurements (Baars et al., 2012). Differences between MISR and CALIOP smoke plume heights are consistent with deeper PBL heights at the time of the CALIOP observation, as PBL is expected to grow further later in the day, and fires might also increase in intensity. We find that PBL height at the location/time of the CALIOP daytime smoke plumes is on an average about 1.4 km higher than for MISR smoke plumes, specifically 2.7 km for CALIOP and 1.3 km for MISR.

Conclusions, Page 16 line 7-9

> We find that CALIOP smoke plume heights are about 1.4-2.2 km higher than MISR smoke plumes, due to a deeper PBL later in the day, possibly more energetic afternoon fires, and CALIOP's greater sensitivity to very thin aerosol layers (Kahn et al., 2008; Flower and Kahn., 2017).

Abstract, page 1 lines 8-11

> A similar pattern is found later in the day (14:00-15:00 LT) with CALIOP, although at higher altitudes (2300 m grassland versus 2000 m tropical forest), as CALIOP typically detects smoke at higher altitudes due to its later overpass time, associated to deeper PBL, possibly more energetic fires, and greater sensitivity to thin aerosol layers.

*4. I strongly suggest comparing your results to what is currently used/assumed in atmospheric aerosol/chemistry models for fire emissions injection heights. Do your results contradict or confirm what is currently used?*

As suggested by the reviewer, we have added the requested information in the conclusions section.

Page 17 line 4-15

> A variety of smoke injection height schemes are used to represent fire emissions over the Amazon, from fire emissions injected below 3 km (Reddington et al., 2016) or into the model-defined PBL (Zhu et al., 2018) to complex plume rise models, in which a significant fraction of emissions are in some conditions injected above 6 km (Freitas et al., 2007). Recent efforts have shown the value of using MISR-derived smoke plume heights to initialise model fire emission injection (Vernon et al., 2018, Zhu et al., 2018). Over the Amazon, Zhu et al., (2018) show that a new injection scheme based on MISR plume-height observations, which included vertical smoke profiles used in this study (Val Martin et al., 2018), provide a better representation of CO observations over the region. With a very narrow swath but sensitivity to sub-visible aerosol, CALIOP tends to sample aerosol layers downwind, providing information complementary to the near-source mapping offered by MISR (Kahn et al., 2008). Thus, observations from both CALIOP and MISR provide a way to study smoke plume heights across the Amazon during the biomass burning season. Ultimately, this information will help improve the representation of biomass burning emissions in Earth system atmospheric models, and should aid our understanding of the feedbacks between droughts, terrestrial ecosystems and atmospheric composition over the region.

*5. Following on from the previous general comments, for modellers it would be extremely useful to have idea of whole vertical distribution of the plumes rather than just median/max plume height. For example, the average percentage of the plume in specified altitude bands. Could this information be estimated from the CALIOP data? Or perhaps this would be unreliable given the large difference between CALIOP and MISR results?*

CALIOP data are not sampled well enough to make the reviewer's suggestion viable. In addition, differences in actual sensitivity between MISR and CALIOP present an additional limitation. The analysis suggested by the reviewer has recently been published in Val Martin et al., (2018). In that work, the Authors present a statistical summary of vertical distribution of smoke (%) by land cover, region and season, from 0 to 8 km at 250 m bins based on MISR stereo-derived smoke plume heights. These profiles are snapshots at the time of the MISR observations (10-11 am LT), but they provide a constraint to initialise fire emission injection heights in climate and atmospheric chemical transport models. Zhu et al., (2018) present an example where this MISR-based injection height scheme is implemented and evaluated within GEOS-Chem, a major atmospheric chemical transport model. As discussed in point 4 above, we added a reference to Val Martin et al (2018) and Zhu et al (2018) in our manuscript.

*Specific comments*

*1. Abstract, L2: Specify the dry season months*

Added as suggested. Note that this is also the primary burning season.

*2. Abstract, L4-6: Sentence not written very clearly "About 60% of smoke plumes are observed during drought years, at the peak month of the burning season (September; 40–50%) and over tropical forest and savanna regions (94%)." Does this mean: 60% of smoke plumes were observed in drought years (relative to non-drought years); 40-50% observed in the peak month of burning season (relative to the other months); and 94% observed over tropical and savannah regions (with the remainder over grassland)?*

We have reworded the Abstract to make these results clearer.

Page 1 lines 4-6

> About 60 % of smoke plumes are observed in drought years, 40-50 % at the peak month of the burning season (September) and 94 % over tropical forest and savanna regions, with respect to the total number of smoke plume observations.

*3. Introduction: Order citations correctly (by year).*

Ordered as suggested.

*4. Introduction, P2, L12-14: Can you include any references for why altitude to which smoke is injected is critical? Perhaps give examples of modelling studies where this has been tested e.g. some of the SAMBBA modelling papers, or observational studies.*

Added references as suggested. Specifically, Jian and Fu, 2014, Archer-Nicholls et al., 2015, Paugam et al., 2016, Zhu et al., 2018.

We have also included information about the SAMMBA modelling studies as follows.

Page 3, lines 6-8

> […] For example, modelling studies during SAMBBA showed the importance of the vertical representation of aerosols from biomass burning over the region (Archer et al., 2015), as biomass burning can modify local weather (Kolusu et al., 2015) and regional climate (Thornhill et al., 2017).

*5. Introduction, P3, L1-3: The Kolusu et al. (2015) and Thornhill et al. (2017) papers are modelling studies not observational (correct this sentence).*

Corrected, as suggested.

*6. Introduction, P3, L9-10: ". . .no analyses yet that seek to quantify the vertical distribution of smoke from fires across the Amazon. . ." Suggest changing to: ". . .quantify the long-term average vertical distribution".*

Changed as suggested.

*7. Sect. 2.1: How are the MISR and MINX vertical resolutions accounted for? Apologies if this explained later in the manuscript.*

The MINX vertical resolution is between 250 and 500 m, depending on observing conditions, and we take it into consideration throughout the study. This is mentioned in page 4 lines 15 and 23.

*8. Sect. 2.6: Why was 0.5x0.5 degrees resolution chosen for CALIOP?*

We ended up having a massive dataset when we compiled raw CALIOP aerosol extinction observations (night and daytime) over the Amazon for 5 months and 7 years. This raw dataset required a long processing time to re-grid them into smoke plumes. So, we first ran a test for one month at 0.1x0.1 and 0.5x0.5 horizontal resolution and decided to process the 7-year observations at 0.5x0.5 as the data were easier and quicker to process and we did not introduce any bias because of the chosen resolution. We clarified this in the manuscript.

Page 7 lines 14-17

> We first grid all CALIOP aerosol extinction profiles classified as smoke (day and night) at a horizontal resolution of $0.5°x0.5°$ over the Amazon region, and a vertical resolution of 250 m, from the surface to 12 km. We chose this horizontal resolution to optimise computing processing time. […]

Page 7 lines 22-25

> To ensure we do not introduce a bias in the CALIOP plume heights due to the 0.5x0.5 horizontal resolution, we also retrieved the smoke plumes for the 2017 burning season at a horizontal resolution of 0.1x0.1, and find no significant differences. For this subset, our 0.5x0.5 method returns 131 plumes, with an average altitude of 3.65 km for the maximum plume heights, whereas the 0.1x0.1 method returns 149 plumes, with an average altitude of 3.74 km.

*9. Sect. 2.6 (P7, L26-35 – P8, L1-5): Nice explanation of the other CALIOP products that have been used by previous studies. This may be helped by a table summarising: the studies (including yours), different products used, region studied etc. Also, is it possible to briefly say what the implications are for using these different products and explain why the specific product and plume height definition were chosen for your study over the others?*

We thank the reviewer again for this suggestion. As mentioned above, we feel that the number of studies over the Amazon is not large enough to justify a table, and our manuscript already contains a large number of tables and figures.

We have reworded section 2.6 to accommodate one comment from reviewer 1 regarding the MISR and CALIOP comparison and why we chose a new approach to derive the smoke plume heights from CALIOP. We hope the new wording/discussion makes this point clearer, and addresses this comment as well.

*10. This sentence from Sect. 3.1: "The majority of the plumes in this record are digitised with blue band retrievals (Table 1). . .", seems to contradict the following two from Sect. 2.2 (or at least have confused me): "This screening leaves a total of 5393 plumes, about 56% of the original database, with 77% and 23% plumes digitised in the red and blue bands, respectively.", and "In our dataset overall, most of plumes are digitised with red band images, as it was the default option for MINX v2–3."*

We refer as "the majority of the plumes in this record" as plumes in 2008, not to the whole dataset as in Section 2.2. We clarified this point in Section 3.1.

*11. Be consistent with use of "PBL"/ "BL".*

Corrected as suggested.

*12. P12, L7 & L17: What are the p-values and at what confidence level is the relationship statistically significant?*

Added as suggested.

*13. P13, L15-16: "Our results suggest that fires during drought periods might significantly degrade regional air quality, as they are associated with low smoke altitude and large aerosol loading.". The finding that drought periods are associated with large aerosol loadings, which substantially degrade regional air quality is consistent with Reddington et al. (2015) (and other studies?). The higher aerosol loadings are likely due to the increases in the number/size of fires (e.g. Aragão et al., 2007; 2014) and subsequent increases in aerosol emissions. However, the potential for lower smoke altitudes in drought years, I'm assuming, is a new finding and should be highlighted/made clearer.*

We thank the reviewer for this comment. We have reworded the manuscript to strengthen this new finding as follows:

Page 13 lines 30-32

> Years with drier conditions have almost a factor of three greater AOD compared with years with wet conditions. Larger aerosol loading in drought periods is likely due to increases in the number and size of fires (e.g., Aragao et al., 2014) and subsequent increases in aerosol emissions.

Abstract lines 18-22

> Consistent with previous studies, the MISR mid-visible aerosol optical depth demonstrates that smoke makes a significant contribution to the total aerosol loading over the Amazon, which in combination with lower injection heights in drought periods, have important implications for air quality. This work highlights the importance of biome type, fire properties and atmospheric and drought conditions for plume dynamics and smoke loading.

*14. P14, L18-19: Would it not be possible to check the PBL height around CALIOP overpass time with MERRA2 data?*

We thank the reviewer for this suggestion. We had already analysed the PBL height at the time of the CALIOP overpass time, but did not include this analysis or show any of the results in the original version. We reworded the manuscript to make this result clearer as discussed in point 3 above.

*15. I'm not sure I understand how Figure 8 demonstrates the following statements: - P12, L7-9: "We find a significant positive relationship between MISR maximum plume heights and MODIS DSI (r=0.7) in tropical forest and savanna fires, with higher maximum plume heights in wet (1000–1100 m) than severe drought conditions (800–900 m) (Figure 8)." - P13, L3-4: "...tropical forest fires inject a larger percentage of smoke plumes into the FT in wet than extreme-dry conditions (12 versus 20%, Figure 8)". Since I can only see data points for tropical forest in "Extreme- Severe" and "Mild Moderate" conditions, with one point in "Normal" and none in "Wetter than Normal".*

We have reworded the text to make our results clearer.

Page 12 Lines 23-25

We find a significant positive relationship between MISR maximum plume heights and MODIS DSI (r=0.7; p<0.01) in tropical forest and savanna fires, with higher maximum plume heights in normal and/or wetter than normal (1000-1100 m) than severe drought conditions (750-900 m) (Figure 8).

Page 13 lines 25-27

Note that in Figure 8 (right bottom), we present the data only subdivided by MODIS DSI and biome, regardless of the year, as in the rest of the panels in Figure 8.

*Tables & Figures*
*1. Table 1: suggest adding a "total" row in the table.*
Added as suggested.

*2. I suggest adding a table to summarise smoke plume heights for the main biomes (could also add drought/non-drought year averages) to compliment Figure 10. So that readers can get this info quicker than reading it off a figure.*
We have most of this information in Table S2 (former Table S1).

*3. Figure 2: I suggest adding one or two figures (perhaps to supplementary) to show the month and/or year the plume occurred e.g. with different colours.*
We thank the reviewer for this suggestion. Showing the smoke plumes per year and month was originally in our plan. However, we tried to recreate the figure using different shapes and colours per month and year but it was hard to make a clear map to show the different locations due to the amount of data points in the 8-year climatology. We decided to present only the main figure with all smoke plume locations in black. In most respects, this information can be gleaned from the plots in Supplemental Material.

*4. Figure 10: If the differences between Day and Night CALIOP Median Plumes are not significant then is it worth just combining these here (keeping the separation in the previous figure)? It is really the difference between the CALIOP and MISR estimations of plume heights that is the significant result.*
We thank the reviewer for this suggestion. We have modified the figure and reworded the manuscript accordingly.

Page 14 Lines 23-28

Figure 10 summarises the median and maximum heights for the CALIOP smoke plumes per biome, season and wet/dry years. Night-time plume heights are on average 250 m higher than daytime plume heights (Figure 9). Differences between day and night-time CALIOP observations have been attributed in the past to a low bias in the daytime retrievals due to noise from scattered solar radiation (e.g., Winker et al., 2009, Huang et al., 2015). Therefore, our difference in day and night-time CALIOP plume heights might result from differences in data quality rather

than reflecting smoke diurnal variability. We combine day and night-time CALIOP observations in Figure 10 and include the MISR plume heights for comparison.

---

## Author Comment (AC3) · 8 Jan 2019

**Response to reviewer #3**

We thank the anonymous reviewer for her/his thorough evaluation and constructive recommendations for improving this manuscript. Her/his comments (in italics) and our responses are listed below.

*General Comment:*

*In this manuscript, the authors characterize burning biomass plume height over the Amazon using MISR and CALIOP observations. They investigate the effect of FRP, atmospheric stability and drought while considering seasonal and interannual variabilities. This is the first time that such work was performed over the Amazon. The manuscript is well structured and well written. The discussion on the drought is particularly interesting. I would recommend this manuscript for publication in ACP after considering the comments listed below. The important point that need to be addressed in the correction is the definition and the use of FRP that cannot be directly linked to fire intensity (see third comment below).*

*Specific comments:*

*P2 L3: consider including in the text some geographical location of where we should materialize this arc of deforestation.*

Defined as suggested.

Page 2 Line 5-6

> Most of these fires burn in the so-called arc of deforestation, along the eastern and southern borders of the Amazon forest, during the dry season.

*P2 L11-14: consider referencing the review on plume injection height from Paugam et al. 2016 when discussing the effect of plume injection height.*

Cited as suggested.

*P2 L21: fire intensity is a specific metrics in fire science expressed in [W/m] and is not the same as FRP[W] or FRP density [W/m2]. However, when dealing with satellite observation, FRP density is usually related to fire intensity. Your definition of fire intensity should be discussed at this point in the introduction. You use in the remaining of the manuscript FRP as a metric for fire intensity. FRP is an estimate of the total radiant energy emitted by the active surface area of the fire, flaming and smoldering area all included. FRP is probably better defined as a measure of fire activity including size and radiant heat flux (FRP density).*

We thank the reviewer for this clarification, and we agree. As discussed in previous work, ours and others (e.g., Kahn et al., 2007; 2008), FRP tends to be a gross underestimate of dynamical heat flux, which is the quantity of interest for plume-rise calculation. The MODIS FRP product, in particular, is reported in the standard product as MW/pixel, and as a MODIS pixel is ~1 km$^2$ except toward the edges of the swath, this amounts to W/m$^2$. So in response to the reviewer comment, we defined FRP more precisely, as suggested, and clarified its meaning throughout the manuscript. Despite the limitations, FRP is one of very few indications of the energy associated with a fire that can be retrieved with remote sensing. So we use it qualitatively as a proxy for fire intensity, with this understanding.

Page 2 lines 24-26

> Related work also demonstrated the important effect that fire radiative power, i.e., a proxy of fire intensity, and atmospheric conditions have on the initial rise of fire emissions (Freitas et al., 2007; Kahn et al., 2007; Val Martin et al., 2012).

Page 4, lines 7-8

> […] We note that MINX provides FRP values in MW, although they are actually in MW per 1-km pixel, which corresponds to $W/m^2$ except toward the edges of the swath.

*P3 L11: capital letter for Fire Radiative Power (FRP). You could add the MODIS collection version here.*

Corrected as noted. We added the MODIS collection version in the Section 2 (Data and Methods)

Page 4, lines 5-7

> The MODIS reports fire radiate power based on a detection algorithm that uses brightness temperature differences in the 4 um and the 11 um channels (Giglio et al., 2003); this FRP parameter is used as an indicator of fire location and intensity. We use MODIS Collection 6 (Table S1 in SI).

*P3 Section2.1: Consider grouping the paragraph on MISR and MODIS, ie l21 to 28 could be moved to the start of page 4.*

We do not understand well what lines need to be moved and where, as we find difficult to match the reviewer's comment to the submitted version of our manuscript. In any case, we consider that grouping the MISR and MODIS discussion will make a lengthy paragraph and prefer to leave it as it is.

*P3 L32: see comment above on FRP and fire intensity.*

See our response to the earlier reviewer comment on the FRP definition.

*P5 L11: replace "),in" to "), in"*

Replaced as suggested.

*P5 L20-23: consider moving the discussion on the red and blue band in the Supplementary Information. As you showed the added error is negligible compared to the MINX uncertainty. You could just mention it once and refer to the SI for more details.*

We thank the reviewer for this suggestion. We agree that the red/blue band error is small compared to the other uncertainties, but decided to keep the discussion in the methodology (section 2.2) as this discussion only adds a small paragraph (6 lines) to the section.

*P6 L2: consider also mentioning here when you consider an atmosphere stable or not.*

The definition for weak and strong atmospheric stability conditions at the plumes is qualitative, based on the atmosphere stability distribution at the smoke plume locations. This definition is addressed once the plume database has been introduced, after section 3.1.

*P7 L2: "wide range of condition as in MISR". I am not sure I understand why your methodology is ensuring a wide range of condition.*

Clarified as noted.

*P5 paragraph 3: I would move this section after you mention the choice of your horizontal resolution (line 5).*

We do not understand what the reviewer refers here. In all versions of the manuscript, page 5 does not mention any resolution. We assume the reviewer refers to the CALIOP horizontal resolution discussion in page 7. We have clarified the choice of horizontal resolution, as also suggested by reviewer 2

*P5 paragraphs order: Consider rearranging the paragraph order in this page to make it easier for the readers. For example, you mention twice how you define CALIOP plume height. This is only a suggestion: the last paragraph on the definition of CALIOP plume height should come after you first mention how you define plume height (line 8). Then would come the discussion on how you link the plume to fire activity.*

We are not sure what order the reviewer means. In any case, to clarify the choice of the CALIOP horizontal resolution we have reordered some paragraphs within this section and we hope the reviewer finds the discussion easier to follow now.

*P5 L18: why do you expect a bias?*

Because of the coarser grid used to estimate the CALIOP smoke plumes. We have clarified this section as suggested by reviewer 2 to make this point clearer.

*P5 L29: Most Probable Height.*

Corrected as suggested.

*P5 L10: consider mentioning that "those grid cells" are the grid cells of your gridded CALIOP injection height product.*

Mentioned as suggested.

*P5 L10: How do you cluster MODIS Fire pixels? Are you taking the larger cluster or do you sum all fire pixel in the grid?*

We assume the reviewer refers on how we use the MODIS fire pixels to consider active fires within the CALIOP smoke plumes. We sum all fire pixels within the grid and only select those grids with at least 2 fire pixels, as explained in page 7, lines 26-28.

*P5 L12: Are you using the same elevation model than in MISR?*

We use a different elevation model than MISR. For CALIOP, as we use a ~50x50 km grid, we estimate the average terrain elevation within the grid based on the CALIOP digital elevation map (GTOPO30). We added this information in the manuscript.

*P8 L15-16: as mention above, I think this is not brining any added value to the discussion here. Move the discussion on MINX band retrieval in the SI.*

We thank the reviewer for the suggestion. As the distinction between red/blue band retrievals is important here, we keep the discussion in section 2.2, but feel this additional reference is helpful here. It only adds one sentence to the paragraph.

*P9 L5-6: Could you mention how does this stability metric relate to the definition of the stability flag (stable/unstable) defined in Val Martin et al 2010? I might miss the point, but why do you define a new metric as you seem to only define atmosphere as weakly and strongly stratified as in Val Martin et al 2010.*

We qualitatively classify atmospheric stability conditions as strong and weak, based on the atmosphere stability distribution calculated at the plume location and time. The atmospheric conditions at the Amazon and North America plumes are different and we cannot use the same classification used in Val Martin et al., (2010). We have clarified this definition to make clear our classification is qualitatively.

*P9 L8: "a summary of these"*

Corrected.

*P9 L14: your value of FRP contrast with the FRP density values reported by Freitas et al 2007 for the same biomes. Grassland is reported to have an FRP density (3.3 kW/m2) an order of magnitude lower than tropical forest (30-80 kW/m2).*

It is difficult to compare our MODIS averaged FRP values and Freitas et al. (2007) heat fluxes.

In our analysis, we obtain the averaged FRP at the location of many fires in early morning. These fires are subjected to many different burning conditions within a particular biome. Freitas et al., (2007) however report a minimum and maximum heat flux per biome. It is not clear to us how those values are estimated as the Authors do not specify it. In Freitas et al., (2006), the Authors only reference a total energy emission measurement over a forest fire in North America as in agreement with their tropical forest fire heat fluxes (30-80 kW/m2). For grasslands, the Authors report one value (3.3 kW/m2) and mention a lack of observations on that type of biome. Our MODIS FRP over grassland is on an average larger than FRP over tropical forests. Our observation is also consistent to that reported in Val Martin et al., (2010) for grasslands versus dry tropical forest over North America.

*P9 L21: "obscuring the fire emitted 4-micron radiance [...] as well as low radiant emissivity". Consider reformulating this sentence. Why the flame emissivity should alter the FRP retrieval in tropical forest? The FRP formulation relies on the gray body assumption. Flaming combustion (because of soot presence in the flame) is more prone to violate the gray body assumption than smoldering. In case of smoldering fire, vegetation absorption is more likely to alter FRP estimate.*

As FRP is measured remotely, we have no way to identify the occurrence, let alone the cause (e.g., due to soot or vegetation absorption or any other factor) of non-unit emissivity at the wavelengths used to measure MODIS FRP. As such, we list smouldering as an example of where non-unit emissivity tends to occur over a broad part of the observed spectrum.

*P9 L25: you could mention that some simulation studies also work on the impact of atmospheric stability and that this is still an open problem in plume rise parameterization. The plume rise model proposed in Paugam et al 2015 (based on the original work of Freitas et al 2007) was shown to be sensitive to atmospheric stability unlike others existing parameterizations. However, this work was refused for publication in ACP, and despite this publication refusal, results of the same model implemented in GFAS were published in ACP in Remy et al 2017.*

We thank the reviewer for this note. We are sorry to hear about the history of Paugan et al., (2015) ACPD work, and agree about the role of the atmospheric stability, which was also included in our own much simpler diagnostic model (Kahn et al., 2007). We extended the discussion and emphasized that there are still some uncertainties in the role of atmospheric stability in plume rise parameterizations.

*P9 L27: consider reformulating: "and weaker atmospheric stability conditions when low altitudes plumes then to be trapped with the boundary layer".*

We do not understand the reviewer's comment, as the suggested sentence does not make sense grammatically. In any case we have reworded the sentence to make it clearer.

Page 10 lines 11-13

> […]. For instance, Val Martin et al. (2012) showed that, in North America, fires that inject smoke to high altitudes tend to be associated with higher FRP and weaker atmospheric stability conditions than those that inject smoke at low altitudes, in which smoke tends to be trapped within the boundary layer.

*P9 L31: as mentioned above, why not using the same flag as in Val Martin et al 2010 to define the state of the atmosphere.*

Addressed above.

*P9 L32: Figure 4*

Unclear note.

*P10 L12-13: I am not sure this sentence brings much to the discussion. Consider removing it.*

Removed as suggested.

*P10 L20-21: combustion efficiency is probably more related to FRP density than FRP. Active fire area is important in your discussion here and should be mentioned.*

We only use the FRP in our assessment and not the active fire area. As such, we consider that mentioning active fire area is out of scope for our study.

*P10 last paragraph: AOD correlate also to the FRP time integration (= Fire Radiative Energy, FRE), see Pereira et al 2009.*

FRE requires integrating a measure of fire energy flux over time. We have only snapshots with MISR and MODIS, so we would need to introduce modelling of some sort to include FRE in the analysis, as Ichoku and Ellison, (2014) do. This is beyond the scope of the current study. We use FRP only as a qualitative indicator, which seems sufficient here.

*P11 L17: why smaller fire in size require less conservative definition in FT injection?*

Smaller fires tend to be less energetic and have lower injection heights. We think that the definition of 'smoke in the FT' proposed for other studies, in which fires were larger and more energetic, is too conservative for the Amazon.

*P11 L24-27: I found the discussion slightly confusing. Does the height the PBL relates to the strength of the stable layer located just above? I might be wrong this is just a thought. In the presence of a deep PBL, there might have quite a lot of water vapor that could be used by the convective plume to get stronger, get across the stable layer and reach the FT.*

Based on our analysis, we cannot determine if the PBL height is related to the strength of the stable layer above, and we cannot determine whether deeper PBLs are associated with more water vapour that can help plume buoyancy.

*P12 L1: "as discussed above". Mention the section.*

Mentioned as suggested.

*P12 L2-3: "Note that DSI is higher in wetter years." Is this not just the definition?*

This comment is to remind the reader how the MODIS DSI is defined, as this is not necessarily intuitive (i.e., the "drought" index is *higher* in wet years…).

*P12 L5: than in severe drought condition.*

Corrected as suggested.

*P12 L23-30: I found the discussion difficult to read. If I well understood your point is that drought effects are correlated with the biome of their geographical location. Drought between 2005 and 2007 move from one biome to another. Could you just discuss FRP and injection height changes for the two biomes between the two years? Why are you using in this discussion the repartition of all observed plume per biome (Fig S1)?*

We state in the manuscript that the regional location of drought makes one biome burn more promptly than the other, as the spatial distribution of biomes over the Amazon is very well defined. For example, northeastern Amazon is dominated by tropical forest whereas southeastern Amazon is dominated by savannah and grassland. Biome determines the type of fire (e.g., smouldering vs. flaming), and hence, FRP and smoke plume heights. We do not say that drought effects are correlated with the biome of their geographical location.

We thank the reviewer for suggesting that we discuss FRP and injection heights in the two biomes and two years. We already discussed this topic on page 13 lines 7-15. In our discussion, we refer to Figure S1, as we think it is important to show the percentages of fires per biome in each year to support our observation that more fires in tropical forest in 2005 and more savannah/grassland fires occurred in 2007.

*P21 L32: what are the mechanisms that make a PBL deeper in dry year?*

PBL is higher in dry years as the surface is warmer, which increases convective mixing. The PBL properties mentioned here are basic meteorology.

*P13 L13: However, you mentioned that grassland fire might reach higher injection height in dry condition?*

We do not understand the reviewer's comment. In our version of the manuscript (page 13 line 13) discusses MODIS DSI and AOD. In any case, in our manuscript we mention that grassland fires inject more smoke plumes into the FT during extreme dry than wet conditions because these fires are associated with high FRP, which may be sufficient to produce the buoyancy needed to lift smoke directly into the FT.

*P14 L17: According to what your argumentation in section 3.4, regardless of PBL, tropical forest fires plumes are lower in dryer condition. So your point here only applies to grassland fire?*

We do not understand the reviewer's comment. There isn't any discussion about tropical forest and grassland fires, and PBL on page 14. Apologies again but we have a hard time following the reviewer's notes with the versions of the manuscript we have available, including the version submitted for the current review.

*P15 L9: "more opportunity to mix upward". MISR data shows generally a peak injection height near the fire where the convective plume is active (with potentially pyroconvection taking place) and then a downdraft caused by the aerosol loading and the atmospheric stratification. A later updraft is possible on longer time scale for older plume through solar radiation heating (De Laat 2012). I think that the main processes responsible of the differences between plume smoke observed by MISR and CALIOP are changes of atmospheric stability and fire activity which can make the updraft core of the plume stronger, making aerosol spreading at higher altitude. Aerosol that were emitted earlier in the day would not have time to reach higher altitude just by solar heating.*

We thank the reviewer for this comment. We have modified the CALIOP and MISR discussion throughout the manuscript to make the results clearer as also suggested by reviewers 1 and 2. Note that differences in the sensitivity of the two techniques would also contribute to CALIOP detecting thin, elevated aerosol above the contrast features detected by MISR in many cases.

*P26 L9-14: as already mentioned, the discussion on fire intensity would be better related to FRP density rather than FRP.*

We think the reviewer means page 16 (conclusions). We have addressed this above.

---

## Author Comment (AC4) · 8 Jan 2019

**Response to Short Comment**

We thank Juliette Koppel for submitting the reviews to our manuscript. We are glad our manuscript was selected as part of an introductory course in the MS in Earth Environment at University of Wageningen. Our responses to Juliette's comments (*in italics*) are listed below.

*General comments*

*The goal of this research is to quantify the vertical distribution of fire smoke across the Amazon and to identify the key factors that control the plume height and rise. In order to achieve this goal, the smoke plume height and its variability will be characterized and the influences of different biome types, fire intensity, local atmospheric conditions and regional drought on smoke height will be studied. The climatology of 2005-2012 is limited for the burning seasons (July – November) and retrieved from space-borne observations from MISR and CALIOP. For all biomes there is a plume height seasonal cycle and also for all biomes most smoke is located below 2 km. No clear relationship is found between drought conditions and fire radiative power. MISR and CALIOP show contradicting results regarding smoke plume heights and DSI, but CALIOP systematically detects higher smoke plumes than MISR. This work highlights the importance of biome type, fire properties and atmospheric conditions for plume dynamics, as well as the effect of drought conditions on smoke loading. The study demonstrates that combined observations of MISR and CALIOP allows for better constraints on the vertical distribution of smoke from biomass burning over the Amazon.*

*What is new in this paper is that there has not yet been any research on the vertical distribution of smoke plumes in the Amazon and also no research has yet been done on the key factors that influence the vertical distribution of fires. This research is of importance because of the great impact of Amazon fires on global biomass burning emissions. These emissions have a large influence on air quality, atmospheric composition, climate and ecosystem health. Therefore, it is necessary to gain a better insight in the vertical distribution of fires and the key factors influencing this process.*

*In my opinion, the paper is written very clear and has a good structure. The introduction is very strong, including societal significance, previous research, the reason of the study area, the gap in research and good funnelling. In general, in the results/discussion section the results that are found are almost all compared with previous studies and explained well. The overall text is easy to read and written in a nice way so that the attention keeps to be drawn to reading the paper.*

*I think this paper fits well to the scope of the journal. The study is about smoke plumes present in the Earth's atmosphere and the underlying physical processes. One of the main research activities of the journal is Remote Sensing, which is in this paper is present in the method because of the use of MISR and CALIOP.*

*However, there are some sections in the paper that need to be revised in order to have this paper published. These adjustments are needed especially in regard to the methods of both MISR and CALIOP, the added value of using both MISR and CALIOP, the importance of land-management policies and some other minor aspects which I will elaborate on later in the review.*

We thank Juliette for these valuable comments. We have addressed the major and minor comments below in a point-by-point basis.

*Major arguments*

*1) MISR and MODIS are both aboard on the NASA Terra satellite, which crosses the equator between 10:00 and 11:00 a.m. local time. This means that observations of smoke plumes will only be available for this time step every day. In this research also the smoke plume heights are related to boundary layer height and atmospheric stability. Specifically, this is done in the results/discussion section, page 9 line 25-34 and page 10 line 1-6. In principle, stable boundary layer conditions occur when θ(K)/Z(km) > 0 and unstable boundary layer conditions occur when θ(K)/Z(km) < 0 (Vilà J., 2017-2018) . In the results and discussion section of this paper an atmospheric stability of 4 K/km is designated as strong, see page 9 lines 33-34. But on what are these values based? All the MISR smoke plumes are categorized as having this weak or strong stability and results (further elaborated in the paragraph below) are based on this. The results can be doubted, since no explanation is given for the criteria values of atmospheric stability and the values thus cannot be validated.*

We have no direct measurements of near-surface atmospheric stability. Model results vary enormously, and must be considered qualitative. As such, we make a reasonable division between low-stability (i.e., small lapse-rate cases) and higher-stability (i.e., larger lapse-rate cases) for the purpose of assessing qualitative differences between these limiting regimes. Our classification is based on the atmospheric stability estimated at the location of the fire plumes over the Amazon, which ranges from -3 to 23 K/km (page 10 line 17). We define the cut-offs in order to have a good representation of data within the two classifications.

To make this point clearer we modify the text as:

Page 10 lines 15-16

> To analyse the influence of atmospheric stability over Amazon fires qualitatively, we divide our plume dataset into two groups that we define as having weak and strong atmospheric stability conditions based on MERRA-2 reanalysis.

On a side note, we kindly ask Juliette and the students in the MS in Earth Environment at University of Wageningen not to reference class notes (e.g. Vilà J., 2017-2018) in future published reviews. Readers outside University of Wageningen do not have access to that material.

*Figure 4 shows the vertical distribution of MISR plume height retrievals, classified under the weak and strong stability categories that are designated here. In lines 2-6, page 10 it is stated that "Our comparison supports previous observations that plumes under weak atmospheric conditions tend to inject smoke to higher altitudes than those experiencing strong stability, with average maximum plume heights of 1150 m and 654 m, respectively." It is also stated that same patterns are found for median and average plume heights. Another statement is that weak stability conditions are associated with deeper boundary layers than strong stability conditions, but it is also stated that this is not even shown. So, first of all, when the categories for weak and strong stability are not appropriately defined, this will cause non appropriate values for the percentage of plumes per category (presented on page 9 line 34 and in figure 4) and maximum, median and average plume heights per category as well (presented on page 10, lines 3-4. Second of all, since it is not even shown that deeper boundary layer heights are associated with weak stability conditions, this statement "Weaker atmospheric stability conditions are also associated with deeper PBLs (~1500 m) than strong stability conditions (~1200 m)." can't be made. On top of that, this very same*

*statement is also a conclusion that is based on the weak/strong stability categories, so when these categories are not defined right, this statement might not even be true.*

The PBL properties cited here are basic meteorology, common knowledge in the field. A detailed discussion of the relationship between PBL stability and MISR-observed plume heights in particular is contained in Val Martin et al., (2010; 2012), which are cited in the current paper.

*Furthermore, the MISR observations are only taken in the morning (10:00-11:00 local time) and thus all the conclusions regarding MISR observations that are made only gives us information for this time step. Since the boundary layer processes and height and atmospheric stability changes a lot during the day (Vilà J., 2017-2018), this time step might not be very representative. Information about the changing boundary layer processes during the day is missing in this paper, where I think it is necessary to include this specifically in the discussion section, page 9 lines 25-24. Also for the conclusions I think it should be stated clearly that this only accounts for the specific time step of (10:00-11:00) and cannot be generalized for the day. In order to be able to test what the effect of changing atmospheric conditions during the day on plume height is, it is necessary to model (with for example model Daysmoke, Liu Y., et al 2010) the hourly PBL height and 6-hourly potential temperature profiles (obtained in this study) against the vertical distribution of smoke plumes.*

The limitation of MISR diurnal sampling is already mentioned in several places in the paper, including the conclusions. Modelling the diurnal cycle would be worth doing, but it is beyond the scope of the current paper.

*2) In the paper it is stated at page 14, lines 5-7, that the initial objective of this research was to compare data from MISR with CALIOP. However, in the paper of Kahn et al., 2008 it is already stated that MISR and CALIOP observations are in fact complementary. Since this is known on beforehand and is mostly due to the properties of both instruments, I don't understand how the authors came to this initial objective. On top of that, in the abstract of the paper, page 1 lines 20-21, it is said that combined observations of MISR and CALIOP allows for better constraints on the vertical distribution of smoke from biomass burning over the Amazon. However, most conclusions in this research are based on the MISR data.*

Our initial aim was to compare smoke plumes observed from both instruments on a plume-by-plume basis, to study the diurnal variability of smoke heights over the Amazon. We developed a new approach to estimate smoke heights on a single-plume basis from CALIOP, and considered a long-term record of observations (7 years). However, despite our efforts, differences in swath widths and sampling times complicate the interpretation of this comparison (page 8 lines 16-23).

Kahn et al., (2008) points out that MISR provides near-source constraints on aerosol plume vertical distributions, whereas in general, CALIOP offers more regional constraints. The current study compares CALIOP and MISR plume-height data on a regional basis, which is both appropriate and useful. As also suggested by reviewers 1 and 2, we clarified this point throughout the manuscript.

*At page 7, lines 2-4, it is mentioned that for CALIOP, both day and night observations will be analysed, to allow a better comparison with the smoke plumes of MISR. But it is already known that comparison of observations of both instruments is not appropriate, and a cause of that is the difference in sampling time. This difference makes it even harder to compare data,*

*because not the same smoke plumes are observed. This is also mentioned in the paper at page 14, lines 5-10. In the results section of the CALIOP smoke plume observations, it is found that the years with highest or lowest number of plumes are the same as observed by MISR and also the peak and biome type with highest biomass burning agree with MISR, page 13 lines 21-24. The only difference in smoke plume heights between MISR and CALIOP were that CALIOP observes smoke at systematically higher altitudes than MISR, stated at page 14 line 31, but this is also already found in previous studies. So for these results, CALIOP has no added value. Also, at line 25 page 14, it is stated that Huang et al., 2015 found the same smoke plume height values over the Amazon. Even though the method of Huang et al., 2015 is different, AOD is calculated for the whole Amazon area, while in this paper the AOD is calculated for individual plumes associated with active fires, no new information is found in this research. Maybe even Huang's results could have been used, because it could have been known that the individual plumes of CALIOP cannot be compared with MISR, so there is no added value in deriving them.*

We disagree with the reviewer here. To our knowledge our study is the first to compare MISR and CALIOP on a plume-by-plume basis over the Amazon. As discussed above, despite having a large sample of plumes in both cases, there were serious limitation to this comparison that we highlighted in the manuscript. Then, to provide context for the MISR observations, we compare them with regional results from CALIOP. One would not expect the two to be identical; the similarities and differences contain important information about both the respective measurement techniques and the regional behaviour of smoke plumes in the Amazon. The MISR data adds considerably to the work of Huang et al., (2015), which used only CALIOP data, and the fact that they reach similar conclusions in many respects adds rather than detracts from the value of analysing this independent dataset.

*So it should be stated more clearly in the methods section of paper, why both instruments are being used in this research and in the results/discussion or conclusions section of the paper, what the additional value is of using both MISR and CALIOP instruments and not just MISR.*

We have revised carefully the MISR-CALIOP comparison throughout the manuscript, as suggested also by reviewers 1 and 2.

*3) In the introduction at page 2 line 4, it is stated that land-management policies cause significant variability in (not mentioned clearly) the spatial variation of fires. After this, in the methods section at page 5 lines 15-16, it is also indicated that one of the years from the climatology (2006) is a year when land-management policies measured limited deforestation. Finally, in the conclusions section at page 16 lines 17-19, the paper states that strong land-management policies can become crucial for the Amazon in controlling fires with changing future climate conditions. Apparently, land-management policies are of importance regarding this research. However, even though one year of adjusted land-management policy is included in the climatology, nothing is mentioned about this in the results/discussion or in the conclusions section. This feels like a missed opportunity, because even though it is only one year in the climatology and maybe nothing significant is found, in the introduction, methods and at the end of the conclusion this research implies that land-management policies could influence biomass burning. Because of this I think this research should include some results or discussion points about this year in the research.*

We mention the land-management policies to inform the reader about specific factors that may affect the number of fires and/or their distribution across the Amazon. However, we do not

analyse the influence of land-management policies on biomass burning as it is out of the scope of our manuscript, and this topic has been covered extensively in the referenced literature (e.g., Nepstad et al., 2005, Aragao et al., 2010 and 2014, Reddington et al., 2015).

*Minor arguments*

*Page 1, abstract/methods: It is nowhere explained why the dataset of MINX is 2005-2012 but the dataset of CALIOP is 2006-2012. CALIOP was launched in 2006, so data of 2005 are impossible to obtain, but why does MINX also includes 2005 in the dataset? Please explain this in the methods.*

The digitalization of MISR smoke plumes is time consuming and requires a huge effort. For this work, we made use of all the smoke plume datasets that had been digitised over the Amazon prior to the focused effort for the current paper (2006, 2007 and 2008). To extend the record to a climatology we added 5 more years. We included 2005 as it was a year with severe drought as 2007 and 2010, and having three years to study the influence of dry conditions on smoke plume heights strengthens the conclusions of this work.

*Page 2, line 4: It is stated that significant variability exists. But it doesn't say between what aspects significant variability exists, so please indicate this more clearly in the text.*

We clarify in the text that significant variability refers to fires and note to the reviewer that all references on line 8 address this point in detail.

*Page 3, lines 15-18: At the end of the introduction the objectives are mentioned. However what is missing here is the influence of land cover/biome type, because that is also studied in the paper. Please include this in the objectives.*

Added as suggested.

*Page 4, lines 13-14: The paper states that a user has to digitise the boundaries of the plume and indicate the direction of the smoke transport. How this should be done however, is not given in the paper. In order to be able to repeat the method I think it is necessary to indicate more clearly how the user should do this, or refer to a paper where this is done.*

The procedure is described in great detail in Nelson et al. (2013), which is cited on page 4 line 24.

*Page 5, lines 10-11: The best estimate maximum and median smoke plume heights are used, but it is not stated how these values are derived. In the paper of Martin M. V., et al 2010, the generation of these values is explained, but is it the same as for this study? And why are these two specific height definitions used and not the other ones that are given by MINX? Please explain this choice.*

These are the same metrics as used in previous studies. They are the main ones produced by MINX, and are derived as described in Nelson et al. (2013), which is cited on page 5 line 19.

*Page 5, lines 11-12: Smoke plumes are categorised with quality retrieval flags, but it is not explained how these categories are derived. The quality retrieval flags determine which plumes are taken into account for the climatology and which are not, so this could affect the total number of observations and it is important to have the right criteria for when a smoke*

*plume should be qualified as good or bad. Thus it is important to be transparent about these quality retrieval flags, so please explain how these are derived.*

This is explained in Nelson et al., (2013) which is cited along the paper, specifically in Section 2.2, and we consider that it is not necessary to be repeated here.

*Page 5, lines 23-25: In the paper it is said that the 60m difference in smoke plume heights between red and blue band retrievals can be neglected, because it is lower than the MINX uncertainty of 250 m. However, when this difference is not negligible this might influence the results because not all observations are retrieved with red and blue band, some only with blue or red band. So also for this, it is important to explain clearly why this difference can be neglected and to add a reference for the MINX uncertainty.*

Nelson et al., (2013) describes the underlying technique, addressing all the related questions in great detail. As such, it is appropriate to reference that paper rather than duplicate it.

*Page 7, lines 14-15: Only the grid cells that contain at least two MODIS fire pixels are associated with active fires, at 80*

We do not understand what the reviewer means.

*Page 7, lines 22-24: To ensure there is no bias in the 0.5x0.5 horizontal resolution, a 0.1x0.1 horizontal resolution for 2017 is obtained and it is stated that there are no significant differences. But it is not stated clearly between what the differences are, please indicate this clearly.*

We have clarified the selection of CALIOP horizontal resolution, as suggested also by reviewers 2 and 3. In any case, we discuss in the manuscript that there is no important bias with respect of the number of plumes and estimated altitude. This is clearly explained in page 7 lines 22-25.

*Page 15/16, conclusion and summary: In my opinion there is not enough of a retrospect towards the reason of why this research has been of importance for the Amazon area. This is very well explained in the introduction and I think it would strengthen the conclusion section and the recommendation for further research, so please elaborate on this is the conclusion section.*

As suggested by reviewer 2, we have made the importance of our findings clearer in the conclusion.

*Minor issues*

*Page 2, line 14: There seems to be a missing reference after the sentence: "The altitude...environmental impact", please include the source.*

Included as suggested.

*Page 4, line 5: In this sentence there is referred to Kahn and Gaitley, 2015. However this reference is not given in the references section, please include this source.*

We have added the reference Kahn and Gaitley (2015) in the references section.

*Page 4, line 24-33: This paragraph is about the limitation of the instruments and might be better for the discussion.*

We thank the suggestion. However, we consider that the discussion of instrument limitations fits well within the methodology.

*Page 4, line 33: There seems to be a missing reference after the sentence: "In contrast...smoke layers", please include the source.*

Referenced as suggested.

*Page 9, line 10: The word "of" is missing before the word "these".*

Corrected, as suggested.

*Page 13, line 4: The word "swallower" should be the word "shallower".*

Corrected, as suggested.

*Page 21, Table 2: Underneath the table there is some additional information where is referred to in the table with an "a" and a "b". However underneath the table there are two "a" and no "b", please change this.*

Corrected, as suggested.

*Page 23, Figure 2: The time series that the figure is given for is not mentioned in the caption, please include this.*

We do not understand what the reviewer means. Figure 2 shows the MISR plume locations over the Amazon domain, without a time series.

*Page 26, Figure 7: For MODIS FRP for the years 2007 and 2009 very high values are found, but nothing is said about this in the results. Also in this figure I don't really understand the necessary of putting the median value also in a number at the top of each boxplot, because it is already indicated inside the boxplot self. If there is no other reason behind putting this numbers here, then please remove them.*

We assume the reviewer refers as 'very high values' to the averages and 67 and 90 percentiles in 2007 and 2009. The text discusses the annual media averages and percentiles are influenced by outliers, as she should know.

We decided to keep the median and number of observations on the top of the boxplots, as it helps the reader easily extract this information from the figure.

*Page 27, Figure 8:*
*The symbols that are used for the years are hard to distinguish and difficult to interpret. Please use other symbols, or make them bigger, or find another way to indicate years.*

We thank the suggestion. We tried to format the symbols in many other ways and that is the setting that we consider clearest. As the reviewer may see, the symbols also include the uncertainty within the annual media, and making the symbols bigger will cover the uncertainty bars in some cases.

To make the figure clearer, we added information to the caption.

"Relationship between MODIS DSI at the location of the plumes and MISR maximum plume height, MODIS FRP and MISR AOD annually averaged, for tropical forest (green), savanna (blue) and grassland (red). Symbols represent the annual average and bars the standard error of the mean. Regression lines are weighted by the number of plumes in each year; relationships with absolute r<0.4 are plotted in dashed lines. Also included percentage of smoke plumes in the FT in each biome and by drought condition. Bar plots indicate the average of [Median Plume--PBL Height]> 0.5 km (light colour) and [Maximum Plume--PBL Height]> 0.25 km (dark colour), based on MERRA-2 PBL heights (see see text for explanation)."